# Improving Non-Transferable Representation Learning by Harnessing Content and Style

**Ziming Hong**[1]  **Zhenyi Wang**[2]  **Li Shen**[3]  **Yu Yao**[4,5]  **Zhuo Huang**[1]
**Shiming Chen**[5]  **Chuanwu Yang**[6]  **Mingming Gong**[5,7]  **Tongliang Liu**[1]*

[1]Sydney AI Centre, The University of Sydney  [2]University of Maryland, College Park  [3]JD Explore Academy
[4]Carnegie Mellon University  [5]Mohamed bin Zayed University of Artificial Intelligence
[6]Huazhong University of Science and Technology  [7]The University of Melbourne

## Abstract

Non-transferable learning (NTL) aims to restrict the generalization of models toward the target domain(s). To this end, existing works learn non-transferable representations by reducing statistical dependence between the source and target domain. However, such statistical methods essentially neglect to distinguish between *styles* and *contents*, leading them to inadvertently fit (i) spurious correlation between *styles* and *labels*, and (ii) fake independence between *contents* and *labels*. Consequently, their performance will be limited when natural distribution shifts occur or malicious intervention is imposed. In this paper, we propose a novel method (dubbed as H-NTL) to understand and advance the NTL problem by introducing a causal model to separately model *content* and *style* as two latent factors, based on which we disentangle and harness them as guidances for learning non-transferable representations with intrinsically causal relationships. Specifically, to avoid fitting spurious correlation and fake independence, we propose a variational inference framework to disentangle the naturally mixed *content factors* and *style factors* under our causal model. Subsequently, based on dual-path knowledge distillation, we harness the disentangled two *factors* as guidances for non-transferable representation learning: (i) we constraint the source domain representations to fit *content factors* (which are the intrinsic cause of *labels*), and (ii) we enforce that the target domain representations fit *style factors* which barely can predict labels. As a result, the learned feature representations follow optimal untransferability toward the target domain and minimal negative influence on the source domain, thus enabling better NTL performance. Empirically, the proposed H-NTL significantly outperforms competing methods by a large margin.

## 1 Introduction

Non-transferable learning (NTL) (Wang et al., 2022b) was proposed as a core technology in intellectual property (IP) protection (Zhang et al., 2018; Chakraborty et al., 2020; Le Merrer et al., 2020; Wang et al., 2024) and controllable artificial intelligence (Zhu et al., 2023; Yang et al., 2023; Li et al., 2023). The critical task in NTL is to learn feature representations (i.e., non-transferable representations) that can maintain source domain performance but cannot perform well on an observed target domain (target-specified NTL task) or any unobserved target domains (source-only NTL task).

Existing NTL methods try to resist the transferability of learned representations by reducing statistical dependence between source domains and target domains (Wang et al., 2022b; Zhu et al., 2023; Zeng & Lu, 2022). In general, they impose two relaxation regularization terms on standard supervised learning: (i) relaxing statistical dependence between "target domain representations" and "target domain labels", (ii) relaxing statistical dependence between "target domain representations" and "source domain representations". However, since such statistical methods are essentially difficult to distinguish the naturally entangled *contents* and *styles* in both source and target domain data, they easily inadvertently fit (i) **spurious correlation** between *styles* and *labels*, and (ii) **fake independence** between *contents* and *labels*.

For intuitive explanation, as shown in Fig. 1 (a), we introduce an NTL example that aims at banning knowledge transferring from the wildlife park (source domain) to the zoo (target domain). We first

---

*Correspondence to Tongliang Liu (tongliang.liu@sydney.edu.au)

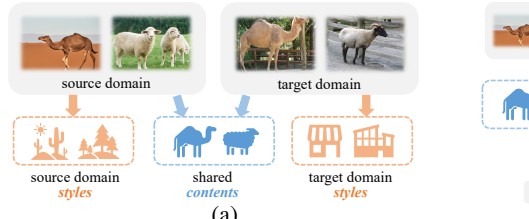 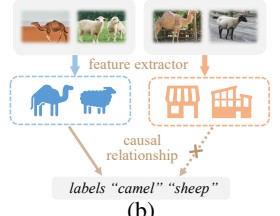 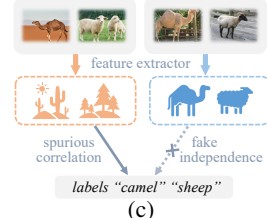

(a)             (b)             (c)

Figure 1: An exmaple: a wildlife park uses *camel* and *sheep* images captured in the wild to train a classification model which cannot be used by the zoo. **(a)** Source and target domain share the same *content factors* (i.e., "*camel*" and "*sheep*") but different *style factors* (i.e., "*natural wild*" versus "*artificial zoo*"). **(b)** Ideal NTL mechanism. **(c)** Existing statistical methods fit **spurious correlation** and **fake independence**.

show the ideal NTL mechanism in line with human consciousness and then the practical situation in using statistical-dependence-relaxation methods. The **ideal and human-like** NTL mechanism is shown in Fig. 1 (b). An NTL model should learn representations of *camel* and *sheep* in the source domain (i.e., *contents*) to implement correct recognition. To limit knowledge transferring, the NTL model should mine the difference between two domains, which naturally and reasonably leads to learning *zoo environment* in the target domain (i.e., *styles*). Such target-specified styles barely contain predictive information and negatively impact source domain recognition. **However, in practical situation, the ideal NTL is destroyed by *nuisance factors*** (Zhang et al., 2022; 2020) (e.g., *desert* and *grassland*, which belong to *styles* and are co-occurrent with *camel* and *sheep* in the wild, respectively). As shown in Fig. 1 (c), since lacking the ability to distinguish the naturally entangled *contents* and *styles*, a statistical-based method will be misled to learn *desert/grassland* in the source domain and *camel/sheep* in the target domain to satisfy the non-transferable requirements. Indeed, knowledge cannot be transferred to the target domain in this case. However, the NTL model **(i)** fits the **spurious correlation** from *styles* to *labels* in the source domain (i.e., *desert* to *label "camel"*, *grassland* to *label "sheep"*) and **(ii)** models the independence from *contents* to *labels* in the target domain (they should have causal relationships, and thus, we call it as **fake independence**). Consequently, the learned non-transferable representations are fragile and sensitive to small distribution shifts (from a training set to a testing set). Thus, both the source domain performance maintenance and target domain degradation will be limited. More seriously, when malicious attackers impose interventions (e.g., domain masquerade attack), such statistical-based methods may not work well since they only fit the superficially statistical relationships in the training data[1].

In this paper, we propose a novel method to address the aforementioned problems by **H**arnessing *content* and *style* in **N**on-**T**ransferable representation **L**earning, dubbed as **H-NTL**. We use a causal model (Peters et al., 2017; Li & Chu, 2023) to formalize the data generation process and mathematically understand the non-transferable mechanism with optimal untransferability. In order to avoid being misled to fit spurious correlation and fake independence, we distinguish the unobservable and naturally mixed *contents* and *styles* by separately modeling them in our causal model. As shown in Fig. 2, we decompose an instance $X$ into two latent factors: (i) **content factor** $C$, which corresponds to the intrinsic, class-related information that is a cause of the label $Y$ in both the souce and the target domain (e.g., bird, airplane, deer, etc.), and (ii) **style factor** $S$, where all other factors except the content factor $C$ can belong to style factors (such as image background or artificial watermark (Wang et al., 2022b)) and is the cause of domain $D$[2]. The *content factor* $C$ and the *style factor* $S$ are causes of the instance $X$. Besides, we assume the co-occurrence between *contents* and *styles* (which leads to the problems of fitting spurious correlation and fake independence in existing NTL methods) is caused by latent confounders, and thus, we model it by a statistical dependence[3] between the *style factor* $S$ and the *content factor* $C$ in our causal model (i.e., the dashed line in Fig. 2) (Von Kügelgen et al., 2021). Based on the causal model, we propose a novel variational inference framework to disentangle the *content factor* $C$ and *style factor* $S$ from observable data by maximizing an approximated evidence lower-bound (ELBO) (Blei et al., 2017; Yao et al., 2021) of the joint distribution $P(X, Y, D)$.

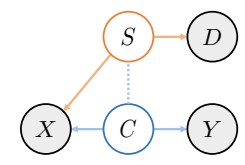

Figure 2: A causal model to reveal the data generation process in NTL. The gray-shaded variables are observable and those unshaded are unobservable. The dashed line indicates the statistical dependence caused by latent confounders.

---

[1]In Appendix A, we present a toy experiment to demonstrate the vulnerability of statistical-based methods.

[2]We further discuss its rationality in NTL and consider another direction (i.e., $D$ causes $S$) in Appendix E.2.

[3]Detailed discussion and assumption of the dependence between $S$ and $C$ is provided in Appendix E.1.

Subsequently, by following the ideal (human-like) NTL mechanism, we harness the disentangled *content factor* $C$ and *style factor* $S$ as guidances for non-transferable representation learning. Based on knowledge distillation (Gou et al., 2021), we treat the NTL model $f_{ntl}$ as a student network and train it by two distillation paths: (i) in the source domain path, we constraint the learned representation to fit *content factor* $C$ (which is the intrinsic cause of *label* $Y$), and (ii) in the target domain path, we enforce the learned representation to fit *style factor* $S$ which barely can predict labels[4]. Therefore, the non-transferable representations learned by $f_{ntl}$ simultaneously approach optimal untransferability toward the target domain and minimal negative influence on source domain recognition, thus enabling better NTL performance. Our contribution can be summarized as follows:

- We identify the problems of fitting (i) *spurious correlation between styles and labels* and (ii) *fake independence between contents and labels* in NTL, which are caused by the statistical dependence between *contents* and *styles*. Such problems lead to the learned non-transferable representations in existing methods become fragile and sensitive to small distribution shifts.
- To address the existing problems, we distinguish the unobservable and naturally mixed *contents* and *styles* by separately modeling them as latent factors in a causal model. Further, we explore causal-driven non-transferable representations with the optimal untransferability, based on which, we propose a simple yet effective method (dubbed as H-NTL) to disentangle and harness the *content factor* and *style factor* for non-transferable representation learning.
- We conduct experiments to evaluate the proposed H-NTL method in both target-specific and source-only NTL tasks. Empirical results across various settings show the superiority of H-NTL.

## 2 RELATED WORK

**Non-transferable learning.** NTL aims to restrict the generalizability of learned models toward a certain domain (target-specified NTL) or all other domains (source-only NTL), which will be widely applied in intellectual property protection (i.e., "ownership verification" and "applicability authorization") and controllable artificial intelligence (Zeng & Lu, 2022; Wang et al., 2022b; Zhu et al., 2023). The most related research fields to the NTL are domain adaptation (DA) (Garg et al., 2022; Saito et al., 2020; Huang et al., 2021; Oza et al., 2023) and domain generalization (DG) (Rame et al., 2022; Zhou et al., 2022; Huang et al., 2023b;a). Specifically, the target-specified NTL is opposite to DA, which aims to enhance the performance on a certain target domain. Similarly, source-only NTL can be seen as an anti-task to single-domain DG. Existing NTL methods try to resist transferability by reducing statistical dependence between representations on source and target domains. Wang et al. (2022b) first propose the NTL task. They design a framework that adds two statistical dependence relaxation terms on standard supervised learning: (i) maximizing the Kullback-Leible (KL) divergence between target domain representations and labels, and (ii) maximizing the maximum mean discrepancy (MMD) between the distribution of source and target domain representations. Zeng & Lu (2022) aim at NTL in natural language processing. They use MMD as well as an auxiliary domain classifier to enlarge the distance of representations from different domains. Further, Zhu et al. (2023) enhance the NTL performance by a domain-weighted MMD strategy. However, such statistical-based methods are easily trapped into fitting spurious correlation and fake independence, which leads to the limitation of both source domain performance maintenance and target domain degradation.

**Causal-inspired representation learning.** Despite the success of statistical learning, they always provide a rather superficial description of limited reality, which means that such success only holds when the experimental conditions are fixed (Schölkopf et al., 2021). In practice, statistical methods are easy to fit spurious correlation from image background to class labels (Zhang et al., 2022; Lv et al., 2022; Ye et al., 2023), which leads to serious performance degradation when facing non-stationary data distribution. Such shortcoming is always reflected in the adversarial vulnerability (Yu et al., 2022; Zhang et al., 2022) and poor generalizability toward unseen domains (Kong et al., 2022; Wang et al., 2022a). Causal-inspired representation learning is a promising approach to address these concerns. It aims to learn representations that are invariant under other changing causal factors, leading to improved performance in the presence of domain shifts (Huang et al., 2022; Mitrovic et al., 2021). Specifically, causal-inspired DA (Kong et al., 2022; Jiang & Veitch, 2022) and DG (Lu et al., 2021; Liu et al., 2021; Chen et al., 2022; Sheth et al., 2022; Chen et al., 2023a) focus on learning invariant representations across different domains to obtain better generalizability. On the contrary, our causal-inspired NTL method aims at learning representations without generalizability in a certain domain(s), which means that the feature representations are "variant" between different domains.

---

[4]We assume a weak relation between $S$ and $C$ in the target domain, which will be empirically verified.

## 3 METHOD

As shown in Fig. 3, we proposed a H-NTL method to handle the non-transferable representation learning problem. In Section 3.1, we first illustrate the ideal non-transferable mechanism based on our causal model. In Section 3.2, we design a variational inference framework to disentangle the unobservable *content factor* $C$ and *style factor* $S$. In Section 3.3, we introduce a dual-path knowledge distillation to harness the disentangled *factors* as guidances to teach a student network to learn ideal non-transferable representations. In Section 3.4, we illustrate the overall training process of H-NTL for target-specified and source-only NTL tasks, respectively.

**Notation.** As shown in Fig. 2, we assume that an instance $X$ in domain $D$ with label $Y$ has two unobservable cause variables: *content factor* $C$ and *style factor* $S$[5]. Particularly, $X$, $Y$, and $D$ are observable variables, and we can access samples from their real joint distribution, namely source domain data $\mathcal{D}_s = \{x_i^s, y_i^s\}_{i=1}^{N_s}$ and target domain data $\mathcal{D}_t = \{x_i^t, y_i^t\}_{i=1}^{N_t}$. We merge these two training sets to a whole dataset $\mathcal{D}$ and represent it as $\mathcal{D} = \mathcal{D}_s \cup \mathcal{D}_t = \{x_i, y_i, d_i\}_{i=1}^{N_s+N_t}$, where $y_i$ is the content label of sample $x_i$ with index $i$, and $d_i$ is the domain label. The goal of NTL is to train a classification model $f_{ntl} : X \to Y$ to degrade performance on target domain $D_t$ and simultaneously maintain performance on source domain $D_s$. Without loss of generality, the $f_{ntl}$ can be splited as a feature extractor $f_e$ and a classifier $f_{cls}$ (i.e., $f_{ntl} = f_{cls} \circ f_e$).

### 3.1 CAUSAL-DRIVEN NON-TRANSFERABLE REPRESENTATION

In this section, to explore the ideal non-transferable mechanism in line with human consciousness (i.e., following intrinsic causal relationship), we provide a causal view to understanding the non-transferable representation learning. Based on the causal model in Fig. 2, the data generation process from *unobservable variables* to *observable variables* can be formalized as:

$$X = f_x(S, C, \epsilon_x), \quad D = f_d(S, \epsilon_d), \quad Y = f_y(C, \epsilon_y) \tag{1}$$

where $\epsilon$ represents the noise term of the corresponding variable, and they are independent of each other and usually can be ignored. Each $f$ represents the corresponding data generation function. Based on Eq. (1), we discuss the non-transferable representations on source/target domain, respectively.

**Source domain representations.** In the source domain, we start from the stable human cognition which relies on the ability of causal reasoning (Zhang et al., 2020; Schölkopf et al., 2021). In order to maintain high classification performance on the source domain, the NTL model $f_{ntl}$ should follow the intrinsic causal relationship from *content factor* $C$ to the *label* $Y$ in Eq. (1). Specifically, the feature extractor $f_e$ in $f_{ntl}$ should model the latent *content factors* (i.e., $f_e(X_s) = C$), and the classifier module $f_{cls}$ is required to establish a mapping from *content factors* to *labels*, i.e., $f_{cls}(C) = Y$.

**Target domain representations.** The next, and more important matter we focus on is representations on the target domain. The crucial problem in non-transferable mechanisms is how to implement untransferability toward the target domain and simultaneously minimize harmful effects on source domain performance (i.e., optimal untransferability). For this purpose, we first explicitly define the properties of optimal untransferability as follows:

**Definition 1** (Optimal Untransferability). *Let $N$ denote a kind of representations across source domain $D_s$ and target domain $D_t$, where $N_s$ and $N_t$ represent source and target domain feature representations, respectively. Let $Y$ denote the classification labels. If $N$ satisfy:*

$$N_t \perp\!\!\!\perp N_s \quad and \quad N_t \perp\!\!\!\perp Y, \tag{2}$$

*we say that the representations $N$ have the optimal untransferability from source domain $D_s$ to target domain $D_t$, where $\perp\!\!\!\perp$ means the statistical independence.*

In Definition 1, the two independent terms simultaneously ensure that (i) the target domain representations contain no information to predict labels (i.e., $N_t \perp\!\!\!\perp Y$) which equals to the untransferability toward the target domain, and (ii) no harmful effect toward source domain prediction (i.e., $N_t \perp\!\!\!\perp N_s$).

**Remark 1.** *We consider the source domain representations mentioned above, and follow the data generation process in Eq. (1). By learning representations $N$ to achieve $N_s = C_s$ on the source domain and $N_t = S$ on the target domain, the $N$ approximately satisfy optimal untransferability.*

---

[5]We use subscript $s$ and $t$ to denote the variable on the source and target domain, respectively. For example, $X_s$ and $C_t$ mean source domain instance and target domain *content factor*, respectively.

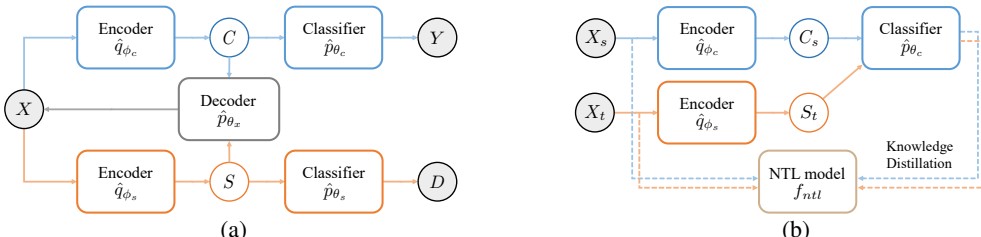

Figure 3: The proposed H-NTL. (a) Disentanglement VAE. (b) Dual-path knowledge distillation.

Intuitively, Remark 1 provides a principle for target domain representation learning, that is, fitting the *style factors* of the target domain (i.e., $f_e(X_t) = S_t$). Under the weak-relation assumption[4], the *style factor* $S$ barely can predict labels although the statistical dependence is existing between $C$ and $S$. Thus, by fitting *style factors* of the target domain, the optimal untransferability toward the target domain can be approximately satisfied under the causal-driven representations in the source domain.

## 3.2 DISENTANGLEMENT UNDER THE CAUSAL MODEL

Although the causal-inspired understanding of the non-transferable mechanism provides a promising direction for practicing this task, both of the *content factor* $C$ and the *style factor* $S$ are unobservable and are naturally entangled in instance $X$. To leverage them for non-transferable representation learning, we have to infer them with the source and target domain datasets $\mathcal{D}_s \cup \mathcal{D}_t$ sampled from the joint distribution $P(X, Y, D)$. In order to simultaneously infer *content factor* $C$ and *style factor* $S$ from sampled data $\mathcal{D}_s \cup \mathcal{D}_t$, according to the causal model in Fig. 2, we factorize the joint distribution $P(X, Y, C, S, D)$ under the Markov condition (Pearl et al., 2000; Yao et al., 2023) as follows:

$$P(X, Y, D, C, S) = P(C, S)P(Y|C, S)P(D|C, S)P(X|C, S), \quad (3)$$

where we see the $C$ and $S$ as a variable set with the joint distribution $P(C, S)$ due to the statistical dependence caused by latent confounders. The Eq. (3) motivates us to extend variational auto-encoder (VAE) (Kingma & Welling, 2013) to infer latent factors, as shown in Fig. 3 (a). Specifically, we separately infer $C$ and $S$ from the input $X$ by using two encoder modules $\hat{q}_{\phi_c}$ and $\hat{q}_{\phi_s}$ to model the posterior distributions $q_{\phi_c}(C|X)$ and $q_{\phi_s}(S|X)$ with learnable parameters $\phi_c$ and $\phi_s$, respectively. In addition, we use a decoder $\hat{p}_{\theta_x}$ to model the distribution $p_{\theta_x}(X|C, S)$ parameterized by $\theta_x$. Moreover, we approximate $P(Y|C, S)$ by assuming that $C$ should contain sufficient information about $Y$ (Yao et al., 2021; Lu et al., 2018), i.e., $P(Y|C, S) = P(Y|C)$, and such assumption is also applicable to $P(D|C, S)$. Therefore, we use two classifiers ($\hat{p}_{\theta_d}$ and $\hat{p}_{\theta_y}$) to model the distributions $p_{\theta_d}(D|S)$ and $p_{\theta_y}(Y|C)$ with learnable parameter $\theta_y$ and $\theta_d$. Totally, the VAE can be presented as a set of network modules: $\{\hat{q}_{\phi_c}, \hat{q}_{\phi_s}, \hat{p}_{\theta_x}, \hat{p}_{\theta_y}, \hat{p}_{\theta_d}\}$.

To learn the parameters $\{\phi_c, \phi_s, \theta_x, \theta_y, \theta_d\}$, we follow the variational inference framework (Blei et al., 2017) to maximize the evidence lower-bound (ELBO) from the sampled observed data $(x, y, d) \in \mathcal{D}$. The $\text{ELBO}(x, y, d)$ is derived as follows (the detailed derivation is shown in Appendix B):

$$\text{ELBO}(x, y, d) = -\text{KL}(q_{\phi_c}(c|x)\|p(C)) - \text{KL}(q_{\phi_s}(s|x)\|p(S)) + \mathbb{E}_{c \sim q_{\phi_c}(c|x)}\left[\log p_{\theta_y}(y|c)\right]$$

$$+ \mathbb{E}_{s \sim q_{\phi_s}(s|x)}\left[\log p_{\theta_d}(d|s)\right] + \mathbb{E}_{c \sim q_{\phi_c}(c|x), s \sim q_{\phi_s}(s|x)}\left[\log p_{\theta_x}(x|c, s)\right], \quad (4)$$

where $\text{KL}(\cdot\|\cdot)$ represents the Kullback-Leibler (KL) divergence. The detailed derivation of Eq. (4) is shown in Appendix B. The first two terms represent the KL divergence between the estimated factors and their prior distributon, i.e., multivariate Gaussian distribution $\mathcal{N}(\mathbf{0}, \mathbf{I})$ (Kingma & Welling, 2013). The third and fourth terms in Eq. (4) are the approximated log-likelihoods of class label prediction from the estimated *content factor* $C$ and domain label prediction from the estimated *style factor* $S$, respectively. The last term can be seen as a reconstruction loss from the estimated latent variables $C$ and $S$ to the input data $X$. We train the VAE by maximizing $\text{ELBO}(x, y, d)$ and thus obtain two encoders $\hat{q}_{\phi_c}(x)$ and $\hat{q}_{\phi_s}(x)$ for disentangling and estimating unobservable *content factor* $C$ and *style factor* $S$, respectively.

Intuitively, by maximizing $\text{ELBO}(x, y, d)$ in Eq. (4), we enforces images with the same label, regardless of their domain, to have similar learned *content factors* $C$. This ensures that the learned $C$ captures class-related information, and disentangle other *style factors* from *contents*. At the same time, we enforces images from different domains to have distinct learned *style factors* $S$, while images from the same domain share similar *style factors*. Thus, the VAE will learn the *style factor* $S$ that captures the stylistic differences across different domains.

### 3.3 DUAL-PATH KNOWLEDGE DISTILLATION

After the disentanglement of *content factor* $C$ and *style factor* $S$, we leverage them as guidances for subsequent non-transferable representation learning. As shown in Fig. 3 (b), in order to satisfy requirements on source and target domains mentioned in Section 3.1, we train the NTL network $f_{ntl}$ in a dual-path knowledge distillation paradigm where the $f_{ntl}$ is seen as a student network.

**Path for source domain.** In the source domain, we follow the causal direction from *content factor* to *label*, i.e., $C \to Y$. It means that the NTL model $f_{ntl} = f_{cls} \circ f_e$ should learn *content factors* through the feature extractor $f_e$, and use the classifier $f_{cls}$ to establish a mapping from feature to label, i.e., $f_{cls}(C) = Y$. The loss of the source domain path can be represented as follows:

$$\mathcal{L}_{kd}^{src} = \mathcal{L}_2(f_{ntl}(x_s), \hat{p}_{\theta_c}(\hat{q}_{\phi_c}(x_s))), \tag{5}$$

where $\mathcal{L}_2$ is the mean square error loss and $x_s$ is a source domain image. In Eq. (5), the learning target is the output logits of the classifier $\hat{p}_{\theta_c}$, with the estimated content $\hat{q}_{\phi_c}(x_s)$ serving as input.

**Path for the target domain.** In the target domain, the aim is to learn representations of *style factors* which barely can predict labels. We present the distillation loss of the target domain path as follows:

$$\mathcal{L}_{kd}^{tgt} = \mathcal{L}_2(f_{ntl}(x_t), \hat{p}_{\theta_c}(\hat{q}_{\phi_s}(x_t))), \tag{6}$$

where $x_t$ is an image sampled from the target domain. Here the learning target is the classifier $\hat{p}_{\theta_c}$ prediction logits of the estimated *style factor* $\hat{q}_{\phi_s}(x_t)$.

**Total knowledge distillation object.** To satisfy both requirements for the source and the target domain, we combine $\mathcal{L}_{kd}^{src}$ and $\mathcal{L}_{kd}^{tgt}$ to the overall knowledge distillation optimization object $\mathcal{L}_{kd}$:

$$\mathcal{L}_{kd} = \mathcal{L}_{kd}^{src} + \lambda_t \mathcal{L}_{kd}^{tgt}, \tag{7}$$

where $\lambda_t$ is a hyperparameter to balance the two loss terms. By minimizing the Eq. (7), the representations learned by the student model $f_{ntl}$ will follow the optimal untransferability to the target domain as well as minimal negative influence toward the source domain. Intuitively, in such a way, given a new source domain image, $f_{ntl}$ aims to predict its content factor, which is directly related to classification. Given a new target domain image, $f_{ntl}$ aims to predict its style factor, which is designed to be the image style that differs across domains and is useless for classification.

### 3.4 OVERALL TRAINING PROCESS OF H-NTL

We present the overall training process of H-NTL in this section. Specifically, the NTL task is categorized into target-specified NTL and source-only NTL according to whether the target domain is known.

**Target-specified NTL task.** As shown in Algorithm 1, we use both the accessible sampled source domain data $\mathcal{D}_s$ and target domain data $\mathcal{D}_t$ to train the VAE and NTL network. For the first $E_{dis}$ epochs, we train the VAE modules through maximizing ELBO in Eq. (4). For the later $E_{kd}$ epochs, we train the NTL network $f_{ntl}$ through minimizing Eq. (7), and particu-

---

**Algorithm 1** Train target-specified H-NTL

1: Training set $\mathcal{D} = \{x_i, y_i, d_i\}_{i=1}^{N_s + N_t}$, Disentanglement epoch $E_{dis}$, KD epoch $E_{kd}$, VAE modules $\{\hat{q}_{\phi_c}, \hat{q}_{\phi_s}, \hat{p}_{\theta_x}, \hat{p}_{\theta_y}, \hat{p}_{\theta_d}\}$ and the NTL network $f_{ntl}$.
2: **for** $i = 1$ to $E_{dis}$ **do**
3:     Train $\{\hat{q}_{\phi_c}, \hat{q}_{\phi_s}, \hat{p}_{\theta_x}, \hat{p}_{\theta_y}, \hat{p}_{\theta_d}\}$ by maximizing ELBO in Eq. (4);
4: **end for**
5: **for** $i = 1$ to $E_{kd}$ **do**
6:     Train $f_{ntl}$ through minimizing Eq. (7);
7: **end for**

---

larly, we freeze all the parameters of VAE in this stage to extract the *content factor* $C_s$ from the coming source domain data $x_s$ and *style factor* $S_t$ from the target domain data $x_t$.

**Source-only NTL task.** In the source-only task, we only access the sampled source domain data $\mathcal{D}_s$. Without loss of generality, we represent the source domain as in-distribution (ID) data and other domains with distribution shifts (including the unseen target domain) as out-of-distribution (OOD) data. Source-only NTL task focuses on degrading the recognition performance for those OOD data with the same *contents* but different *styles* (Wang et al., 2022b). Unlike the domain augmentation method proposed in Wang et al. (2022b) where generative adversarial network (GAN) is used to synthesize OOD data and make the source-only NTL feasible, we achieve this goal by imposing interventions on the *style factor* (i.e., $do(S)$) (Von Kügelgen et al., 2021; Mitrovic et al., 2021). Due to the independence of mechanisms (Peters et al., 2017), $do(S)$ do not impact $P(Y|C)$, which means that manipulating the value of $S$ does not change the *content factor* $C$. Thus, the OOD data obtained by interventions meet the requirements in source-only NTL scenarios. In practice,

Table 1: Comparison on target-specified NTL tasks. For each table cell, the first line shows averaged accuracy $Acc$ (%) with standard deviation, and the second line shows accuracy drop $\Delta$ (behind ↓) and relative drop $\Delta\%$ (in brackets) compared to supervised learning (SL). The best results (except SL) of "*source/target domain performance*" and "*performance difference between domains*" are highlighted in "underline" and "**bold**", respectively.

| Source→ Target | Img Size | SL | | tNTL (Wang et al., 2022b) | | H-NTL (Ours) | |
|---|---|---|---|---|---|---|---|
| | | Source | Target | Source | Target | Source | Target |
| MM→MT | 32 | 94.30 ±0.79 – | 97.47 ±0.40 – | 88.33 ±1.25 ↓ 5.97 (6.33%) | 8.47 ±2.29 ↓ 89.00 (91.31%) | **93.10 ±0.96** ↓ **1.20 (1.27%)** | **9.90 ±0.60** ↓ **87.57 (89.84%)** |
| SN→SD | 32 | 87.47 ±0.55 – | 50.33 ±5.32 – | 81.33 ±1.66 ↓ 6.14 (7.02%) | 10.27 ±0.81 ↓ 40.06 (79.59%) | **88.13 ±1.43** ↓ **-0.66 (-0.75%)** | **9.23 ±1.35** ↓ **41.10 (81.66%)** |
| SD→MT | 32 | 98.23 ±0.06 – | 55.30 ±3.00 – | 89.50 ±0.66 ↓ 8.73 (8.89%) | 9.20 ±1.01 ↓ 46.10 (83.36%) | **97.13 ±0.35** ↓ **1.10 (1.12%)** | **10.97 ±0.83** ↓ **44.33 (80.16%)** |
| C10→S10 | 32 | 81.10 ±0.20 – | 61.60 ±1.32 – | 79.77 ±1.32 ↓ 1.33 (1.64%) | 49.60 ±4.30 ↓ 12.00 (19.48%) | **80.63 ±0.84** ↓ **0.47 (0.58%)** | **28.10 ±4.76** ↓ **33.50 (54.38%)** |
| | 64 | 86.57 ±0.38 – | 67.60 ±0.95 – | 86.53 ±1.61 ↓ 0.04 (0.05%) | 9.93 ±0.90 ↓ 57.67 (85.31%) | **87.60 ±0.26** ↓ **-1.03 (-1.19%)** | **9.63 ±1.50** ↓ **57.97 (85.75%)** |
| VT→VV | 32 | 89.67 ±0.80 – | 22.43 ±3.56 – | 88.47 ±0.21 ↓ 1.20 (1.34%) | 8.27 ±0.81 ↓ 14.16 (63.13%) | **91.73 ±1.12** ↓ **-2.06 (-2.30%)** | **8.07 ±0.71** ↓ **14.36 (64.02%)** |
| | 64 | 93.40 ±0.70 – | 35.60 ±1.56 – | 93.33 ±0.35 ↓ 0.07 (0.07%) | 8.47 ±0.92 ↓ 27.13 (76.21%) | **94.60 ±0.44** ↓ **-1.20 (-1.28%)** | **8.20 ±0.92** ↓ **27.40 (76.97%)** |
| OP→OC | 32 | 65.57 ±1.27 – | 23.57 ±1.01 – | 62.80 ±1.49 ↓ 2.77 (4.22%) | 15.90 ±0.20 ↓ 7.67 (32.54%) | **65.60 ±0.61** ↓ **-0.03 (-0.05%)** | **5.50 ±0.98** ↓ **18.07 (76.67%)** |
| | 64 | 75.60 ±1.47 – | 31.63 ±1.02 – | 74.13 ±0.06 ↓ 1.47 (1.94%) | 22.33 ±0.40 ↓ 9.30 (29.40%) | **76.43 ±0.91** ↓ **-0.83 (-1.10%)** | **6.73 ±0.67** ↓ **24.9 (78.72%)** |

considering the fact that we cannot obtain naturally entangled *content factor C* and *style factor S* from a single source domain and subsequently manipulate the *style factor* value, we perform image style augmentation (Sohn et al., 2020; Berthelot et al., 2019; Cubuk et al., 2020) on source domain data. Image style augmentations (e.g., blurring, sharpness, solarize) do not influence the *contents* but significantly change the image styles, thus satisfying our intervention aims. Refer to Appendix C.4 for detailed style augmentations. We see all augmented images as the target domain, and then we use the accessible source domain and the target domain to train the H-NTL in the same way as target-specified NTL. The full algorithm of source-only H-NTL is shown in Appendix C.5.

## 4 EXPERIMENTS

We conduct a series of experiments to evaluate the effectiveness of our H-NTL, and compare it with supervised learning (SL) and the NTL method proposed by Wang et al. (2022b) (tNTL and sNTL[6]). Our experiments involve three basic digit tasks and three challenging tasks on real-world datasets. The digit tasks contain three random-selected pairs from four digit datasets: *MNIST* (MT) (Deng, 2012), *MNIST-M* (MM) (Ganin et al., 2016), *SVHN* (SN) (Netzer et al., 2011) and *SYN-D* (SD) (Roy et al., 2018). For challenging tasks, we involve *CIFAR10* to *STL10* (Coates et al., 2011) (C10→S10), *VisDA* (Peng et al., 2017) (VT→VV) and *OfficeHome* (Venkateswara et al., 2017) (OP→OC). We resize images to $32 \times 32$ on digit tasks and both $32 \times 32$ and $64 \times 64$ for challenging tasks. We follow the backbones in (Wang et al., 2022b) to conduct all experiments. To evaluate, we present top-1 accuracy ($Acc$) on both source and target domain. Results are reported as average with standard deviation over three independent runs. Additionally, we calculate the averaged accuracy drop ($\Delta = Acc_{sl} - Acc_{ntl}$) and relative drop ($\Delta\% = \Delta / Acc_{sl}$) on source/target domain, respectively. More implementation details (e.g., datasets, network architecture, training details) are given in Appendix C.

### 4.1 EXPERIMENTS ON TARGET-SPECIFIED NTL TASK

**Comparison with baselines.** We compare the proposed H-NTL with baselines (SL and tNTL), and the results are shown in Table 1. H-NTL outperforms tNTL regarding the *difference of source-target domain performance* on all datasets. For MM→MT and SD→MT, tNTL has better target domain performance degradation (91.31% and 83.36%) compared to H-NTL (89.84% and 80.16%), but tNTL also impacts more source domain performance, with the relative drop $\Delta\%$ achieving 6.33%

---

[6]In order to avoid confusion between the method name and the task name, we here use tNTL and sNTL to separately denote their method for target-specified NTL task and source-only NTL task.

Table 2: Comparison on source-only NTL tasks. For each table cell, the first line shows averaged accuracy $Acc$ (%) with standard deviation, and the second line shows accuracy drop $\Delta$ (behind ↓) and relative drop $\Delta\%$ (in brackets) compared to supervised learning (SL). The best results (except SL) of "*source/target domain performance*" and "*performance difference between domains*" are highlighted in "underline" and "**bold**", respectively.

| Source→ Target | Img Size | SL | | sNTL (Wang et al., 2022b) | | H-NTL (Ours) | |
|---|---|---|---|---|---|---|---|
| | | Source | Target | Source | Target | Source | Target |
| MM→MT | 32 | 94.30 ±0.79 – | 97.47 ±0.40 – | 89.67 ±1.24 ↓ 4.63 (4.91%) | 22.93 ±2.37 ↓ 74.54 (76.47%) | **92.10 ±0.85** ↓ **2.20 (2.33%)** | **13.43 ±3.41** ↓ **84.04 (86.22%)** |
| SN→SD | 32 | 87.47 ±0.55 – | 50.33 ±5.32 – | 86.20 ±0.40 ↓ 1.27 (1.45%) | 18.60 ±1.59 ↓ 31.73 (63.04%) | **86.70 ±2.26** ↓ **0.77 (0.88%)** | **10.70 ±1.47** ↓ **39.63 (78.74%)** |
| SD→MT | 32 | 98.23 ±0.06 – | 55.30 ±3.00 – | 96.27 ±0.42 ↓ 1.96 (2.00%) | 14.40 ±3.72 ↓ 40.90 (73.96%) | 95.17 ±1.10 ↓ 3.06 (3.12%) | **10.47 ±0.96** ↓ **44.83 (81.07%)** |
| C10→S10 | 32 | 81.10 ±0.20 – | 61.60 ±1.32 – | 83.13 ±0.81 ↓ -2.03 (-2.50%) | 63.50 ±0.75 ↓ -1.90 (-3.08%) | 76.30 ±2.98 ↓ **4.80 (5.92%)** | **52.70 ±1.90** ↓ **8.90 (14.45%)** |
| | 64 | 86.57 ±0.38 – | 67.60 ±0.95 – | 84.67 ±0.99 ↓ 1.90 (2.19%) | 47.23 ±3.99 ↓ 20.37 (30.13%) | **86.63 ±2.08** ↓ **-0.06 (-0.07%)** | **9.80 ±1.11** ↓ **57.80 (85.50%)** |
| VT→VV | 32 | 89.67 ±0.80 – | 22.43 ±3.56 – | 91.70 ±0.70 ↓ -2.03 (-2.26%) | 17.53 ±1.46 ↓ 4.90 (21.85%) | 91.63 ±1.05 ↓ **-1.96 (-2.19%)** | **9.27 ±1.30** ↓ **13.16 (58.67%)** |
| | 64 | 93.40 ±0.70 – | 35.60 ±1.56 – | 91.93 ±0.93 ↓ 1.47 (1.57%) | 18.37 ±1.00 ↓ 17.23 (48.40%) | **94.93 ±0.57** ↓ **-1.53 (-1.64%)** | **7.80 ±0.35** ↓ **27.80 (78.09%)** |
| OP→OC | 32 | 65.57 ±1.27 – | 23.57 ±1.01 – | 60.73 ±0.45 ↓ 4.84 (7.38%) | 19.13 ±0.50 ↓ 4.44 (18.84%) | **63.73 ±0.12** ↓ **1.84 (2.81%)** | **14.43 ±0.95** ↓ **9.14 (38.78%)** |
| | 64 | 75.60 ±1.47 – | 31.63 ±1.02 – | 71.20 ±0.87 ↓ 4.40 (5.82%) | 26.97 ±0.78 ↓ 4.66 (14.73%) | **71.63 ±1.84** ↓ **3.97 (5.25%)** | **16.00 ±2.55** ↓ **15.63 (49.42%)** |

Figure 4: Results of target-specified NTL on watermarked data, where the "(src)" and "(tgt)" behind each method name denote the accuracy on source and target domain, respectively.

and $8.89\%$, respectively. For challenging tasks with low-resolution images and more categories (e.g., OP→OC and C10→S10 with $32 \times 32$ image size), thanks to the disentanglement of *content factor* and *style factor* and dual-path knowledge distillation, H-NTL can still learn effective non-transferable representation and achieve source domain maintenance and target domain degradation, while tNTL fails in both aspects. Overall, these results demonstrate the effectiveness of the proposed H-NTL in target-specified NTL.

**Target-specified NTL on watermark data.** We also conduct target-specified NTL experiments on watermark data, in which we use pixel-level mask patches as watermarks and add them to the original source domain data to form target domain (Wang et al., 2022b). To test the ability of learning non-transferable representations, we conduct experiments under different patch values. The value reflects the degree of distribution shifts from the original source domain to the target domain with human-added watermarks. It is more difficult for a model to learn non-transferable representations with the patch value decreasing. The results on C10 and S10 are shown in Fig. 4. On either C10 or S10, the target domain performance degradation of tNTL is gradually limited, with the patch value decreasing. In contrast, H-NTL is slightly affected (S10) or even not affected (C10) by the patch value dropping. This is because the disentanglement and dual-path knowledge distillation in H-NTL can better distinguish and guide the $f_{ntl}$ to fit *content factor* on the source domain and *style factor* on the target domain, even only slight distribution shifts existing between source and target domain.

## 4.2 EXPERIMENTS ON SOURCE-ONLY NTL TASK

The results of source-only NTL tasks are shown in Table 2. For all datasets, the proposed H-NTL outperforms sNTL (Wang et al., 2022b) on the evaluation metric of *difference on source-target domain performance*, which reflects the overall advantages of the proposed H-NTL as well as the

Table 3: Ablation Studies. For target-specified NTL and source-only NTL, the best results of "*difference on source-target domain performance*" are highlighted in "**bold**".

| Task | Method | C10→S10 | | VT→VV | | OP→OC | |
|------|--------|---------|---|-------|---|-------|---|
| | | Source | Target | Source | Target | Source | Target |
| – | SL | 86.57 ±0.38 | 67.60 ±0.95 | 93.40 ±0.70 | 35.60 ±1.56 | 75.60 ±1.47 | 31.63 ±1.02 |
| target-specified | H-NTL w/o $C$ | 10.80 ±0.61 ↓ 75.77 (87.52%) | 9.77 ±2.18 ↓ 57.83 (85.55%) | 8.87 ±0.50 ↓ 84.53 (90.50%) | 8.17 ±0.75 ↓ 27.43 (77.05%) | 1.93 ±0.31 ↓ 73.67 (97.45%) | 1.17 ±0.74 ↓ 30.46 (96.30%) |
| | H-NTL w/o $S$ | 87.90 ±0.44 ↓ -1.33 (-1.54%) | 72.40 ±1.30 ↓ -4.80 (-7.10%) | 94.80 ±0.36 ↓ -1.40 (-1.50%) | 36.20 ±0.53 ↓ -0.60 (-1.69%) | 77.60 ±1.82 ↓ -2.00 (-2.65%) | 39.10 ±1.14 ↓ -7.47 (-23.62%) |
| | H-NTL (full) | **87.60 ±0.26** ↓ **-1.03 (-1.19%)** | **9.63 ±1.50** ↓ **57.97 (85.75%)** | **94.60 ±0.44** ↓ **-1.20 (-1.28%)** | **8.20 ±0.92** ↓ **27.40 (76.97%)** | **76.43 ±0.91** ↓ **-0.83 (-1.10%)** | **6.73 ±0.67** ↓ **24.9 (78.72%)** |
| source-only | H-NTL w/o $C$ | 10.80 ±1.01 ↓ 75.77 (87.52%) | 9.87 ±1.21 ↓ 57.73 (85.40%) | 9.17 ±0.67 ↓ 84.23 (90.18%) | 8.60 ±1.45 ↓ 27.00 (75.84%) | 2.40 ±0.17 ↓ 73.20 (96.83%) | 1.40 ±0.52 ↓ 30.23 (95.57%) |
| | H-NTL w/o $S$ | 87.93 ±1.67 ↓ -1.36 (-1.57%) | 66.07 ±2.38 ↓ 1.53 (2.26%) | 95.77 ±0.67 ↓ -2.37 (-2.54%) | 24.23 ±1.68 ↓ 11.37 (31.94%) | 76.93 ±0.42 ↓ -1.33 (-1.76%) | 31.60 ±0.75 ↓ 0.03 (0.09%) |
| | H-NTL (full) | **86.63 ±2.08** ↓ **-0.06 (-0.07%)** | **9.80 ±1.11** ↓ **57.80 (85.50%)** | **94.93 ±0.57** ↓ **-1.53 (-1.64%)** | **7.80 ±0.35** ↓ **27.80 (78.09%)** | **71.63 ±1.84** ↓ **3.97 (5.25%)** | **16.00 ±2.55** ↓ **15.63 (49.42%)** |

effectiveness of our style augmentation strategies for the source-only task. On SD→MT and VT→VV ($32 \times 32$), the source domain accuracies of sNTL slightly exceed our H-NTL ($1.1\%$ and $0.07\%$). However, regarding the degradation of target domain accuracy, H-NTL outperforms sNTL by a large margin ($3.93\%$ and $8.26\%$). For the task of C10→S10 ($32 \times 32$), due to the low resolution of images, both sNTL and H-NTL are hard to learn effective non-transferable representations when only the source domain is available. Owing to the disentanglement, H-NTL can still capture more effective *content factor* and *style factor* and learn better non-transferable representations, with the *difference on source-target domain performance* exceeding sNTL $3.97\%$.

## 4.3 ABLATION STUDIES

In this section, we explore the effectiveness of main components in our H-NTL. We conduct ablation studies on three paired datasets in both target-specified and source-only tasks: C10→S10, VT→VV, and OC→OP, with images resized to $64 \times 64$. As shown in Table 3, without the guidance of *content factors* in the source domain (i.e., H-NTL w/o $C$), both the source and target domain performance will drop to random classification accuracy (either target-specified or source-only task). Without the guidance of *style factors* in target domain (i.e., H-NTL w/o $S$), the source and target domain accuracies are comparable to or even exceed the accuracies of supervised learning (SL). This is because the student NTL network $f_{ntl}$ fits the disentangled *content factors* from the paired source-target domain (in target-specified NTL) or original source and augmented source domain (in source-only NTL), which leads to better generalizability. Only with the full guidance from *content* and *style factors*, i.e., H-NTL (full), the non-transferable representation learning can be successfully achieved.

## 4.4 ADDITIONAL EXPERIMENTS AND ANALYSES

Due to the page limitation, we present additional experiments and analyses in Appendices, which include but are not limited to a toy experiment (Appendix A), additional experiments on high-resolution images (Appendix D.1), IP protection (Appendix D.2), more baseline (Appendix D.3), the influence of KD loss weight (Appendix D.4), the influence of style augmentation number (Appendix D.5), disentanglement analysis (Appendix D.6) and visualization results (Appendix D.7).

## 5 CONCLUSION

In this paper, we propose a H-NTL method to handle the NTL problem. To understand the non-transferable mechanism, we introduce a causal model to formalize the data generation process, in which we separately model the unobservable contents and styles by two latent factors. Then, we propose a variational inference framework to disentangle the naturally mixed *content factor* and *style factor* from observable data. Furthermore, we harness the disentangled two *factors* as guidances for learning non-transferable representations. By dual-path knowledge distillation, we enforce the source domain representations learned by the student model to fit *content factors* and the target domain representations to fit *style factors*. The learned representations follow optimal untransferability toward the target domain and minimal impact on the source domain, thus boosting the NTL performance. Extensive experiments are conducted to validate the effectiveness of H-NTL.

ACKNOWLEDGEMENTS

Li Shen is supported by STI 2030—Major Projects (No. 2021ZD0201405). Ziming Hong is supported by JD Technology Scholarship for Postgraduate Research in Artificial Intelligence No. SC4103. Mingming Gong is supported by ARC DE210101624. Tongliang Liu is partially supported by the following Australian Research Council projects: FT220100318, DP220102121, LP220100527, LP220200949, IC190100031.

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

# APPENDICES

OVERVIEW:

- Appendix A contains a toy experiment to present the vulnerability of statistical methods.
- Appendix B contains full derivation of formulas in the main paper.
- Appendix C contains complementary experimental details.
- Appendix D contains additional experimental results and analyses.
- Appendix E contains discussion of the rationality of our causal model.

## A    TOY EXPERIMENT

In this section, through a toy experiment, we show that statistical methods (e.g., Wang et al. (2022b)) are easily misled to (i) fit **spurious correlation** between *style factors* and *labels*, and (ii) model **fake independence** between *content factors* and *labels*, rather than the intrinsically causal relationship. At the same time, we present a target domain masquerade attack toward these statistical methods by simply imposing intervention on the source domain.

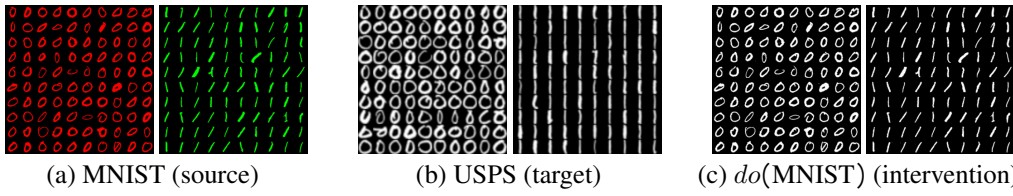

|   (a) MNIST (source)   |   (b) USPS (target)   |   (c) $do$(MNIST) (intervention)   |

Figure 5: A toy NTL experiment from MNIST to USPS.

As shown in Fig. 5, we conduct the toy NTL experiment on digit datasets, where we use MNIST (Deng, 2012) as the source domain and USPS (Hull, 1994) as the target domain. We only use two classes in these digit datasets for the toy experiment, i.e., number "0" and number "1". As shown in Fig. 5 (a), each digit in the source domain has a human-added class-wise color (**red** for number "0" and **green** for number "1"). In the target domain, as shown in Fig. 5 (b), we keep the original data. We randomly selected 400 class-balanced samples from each domain for training and 200 for testing.

Moreover, in the testing phase, we are not satisfied with evaluating these methods by regular samples. As shown in Fig. 5 (c), we intervene the source domain by simply changing the color of digits to white, i.e., $do$(MNIST). In the current task, changing color will not influence the *contents* which are the intrinsical causes of *labels*. What we expect is that an NTL model can fit *content factors* in the source domain and *style factors* in the target domain, thus leading to the ideal and robust NTL mechanism in line with human consciousness. If an NTL model cannot maintain source domain performance under the intervention $do$(MNIST), it means that the model fits (i) **spurious correlation** between *style factors* and *labels*, and (ii) **fake independence** between *content factors* and *labels*, rather than the intrinsically causal relationship. At the same time, such intervention can be seen as a target domain masquerade attack toward these methods.

Based on the above considerations, we train our causal-inspired H-NTL and the statistical-based method proposed by Wang et al. (2022b) (we denote it as tNTL to avoid being confused with the task name). Following Wang et al. (2022b), we use VGG-11 (Simonyan & Zisserman, 2015) pretrained on ImageNet-1K (Deng et al., 2009) as backbones for both H-NTL and tNTL. To evaluate, we present top-1 accuracy ($Acc$) on both the source and target domain. Results are reported as average with standard deviation over three independent runs.

As shown in Table 4, both tNTL and H-NTL reach excellent NTL performance in regular testing set, with the source domain (MNIST) accuracy reaching $100\%$ and the target domain (USPS) performance degrading to random classification. However, facing intervened MNIST data, the statistical-based method (tNTL) failed to implement correct recognition, with the accuracy dropping to $49.17\%$. This phenomenon illustrates that the tNTL fits the **spurious correlation** from *style factors* to *labels* (i.e., **red**→"0" and **green**→"1"), rather than the real *content factors*. Simultaneously, the tNTL relaxes the statistical dependence between *labels* and *content factors* as well as other *style factors* (such as the

Table 4: Results of the toy NTL experiment (MNIST→ USPS). We show averaged accuracy $Acc$ (%) with standard deviation from three independent runs.

| Method | MNIST (source) | USPS (target) | $do$(MNIST) (intervention) |
|---|---|---|---|
| tNTL (Wang et al. (2022b), statistical-based) | 100.00 ±0.00 | 50.83 ±1.04 | 49.17 ±3.33 |
| H-NTL (ours, causal-inspired) | 100.00 ±0.00 | 51.50 ±2.60 | 97.17 ±2.02 |

digital styles) to implement untransferability. In particular, the imposed relaxation between *labels* and *content factors* belongs to **fake independence**. On the contrary, our causal-inspired H-NTL correctly practices the ideal NTL mechanism in line with human consciousness, with the accuracy still reaching 97.17% under intervention.

Overall, this toy experiment shows the vulnerability of statistical-based NTL methods. In practice, fitting **spurious correlation** and **fake independence** will lead to the learned non-transferable representations being fragile and sensitive to small natural or intervention-based domain shifts. This further leads to the limitation on both source domain performance maintenance and target domain performance degradation. Moreover, this also indicates the risk of being attacked by malicious attackers through target domain masquerading.

## B  THE DERIVATION OF ELBO

In this section, we provide a detailed derivation of the evidence lower-bound (ELBO) in Eq. (4) in the main paper. Now, we start with maximizing the log-likelihood $p(x, y, d)$ of each datapoint $(x, y, d)$ from the dataset $\mathcal{D}$. The log-likelihood of the joint distribution can be written as:

$$
\begin{aligned}
\log p(x, y, d) &= \log \int_c \int_s p(x, y, d, c, s) \mathrm{d}c\mathrm{d}s \\
&= \log \int_c \int_s p(x, y, d, c, s) \frac{q_\phi(c, s|x)}{q_\phi(c, s|x)} \mathrm{d}c\mathrm{d}s \\
&= \log \mathbb{E}_{(c,s)\sim q_\phi(c,s|x)} \left[ \frac{p(x, y, d, c, s)}{q_\phi(c, s|x)} \right] \\
&\geq \mathbb{E}_{(c,s)\sim q_\phi(c,s|x)} \left[ \log \frac{p(x, y, d, c, s)}{q_\phi(c, s|x)} \right] := \mathrm{ELBO}(\mathrm{x}, \mathrm{y}, \mathrm{d}).
\end{aligned}
\tag{8}
$$

By applying the factorization in Eq. (3), we have:

$$
\begin{aligned}
\mathrm{ELBO}(x, y, d) &= \mathbb{E}_{(c,s)\sim q_\phi(c,s|x)} \left[ \log \frac{p(c, s)p_{\theta_y}(y|c, s)p_{\theta_d}(d|c, s)p_{\theta_x}(x|c, s)}{q_\phi(c, s|x)} \right]. \\
&= \mathbb{E}_{(c,s)\sim q_\phi(c,s|x)} \left[ \log \frac{p(c, s)}{q_\phi(c, s|x)} \right] + \mathbb{E}_{(c,s)\sim q_\phi(c,s|x)} \left[ \log p_{\theta_y}(y|c, s) \right] \\
&\quad + \mathbb{E}_{(c,s)\sim q_\phi(c,s|x)} \left[ \log p_{\theta_d}(d|c, s) \right] + \mathbb{E}_{(c,s)\sim q_\phi(c,s|x)} \left[ \log p_{\theta_x}(x|c, s) \right].
\end{aligned}
\tag{9}
$$

In practice, to reduce computational costs and to allow our method efficiently infer latent factors, we approximate $P(Y|C, S)$ by assuming that $C$ contains sufficient information about $Y$ (Yao et al., 2021; Lu et al., 2018), i.e., $P(Y|C, S) = P(Y|C)$, and such assumption is also applicable to $P(D|C, S)$. Moreover, we separately infer $C$ and $S$ from only the input $X$ by using two encoder modules $\hat{q}_{\phi_c}$ and $\hat{q}_{\phi_s}$ to model the posterior distributions $q_{\phi_c}(C|X)$ and $q_{\phi_s}(S|X)$, respectively. Due to the fact that the complex image $X$ contains enough information about $C$ and $S$ in real scenarios (Yao et al., 2021; Lu et al., 2018), such assumptions are reasonable and are expected not to have very large approximation error. Therefore, the Eq. (9) can be derived as:

$$
\begin{aligned}
\mathrm{ELBO}(x, y, d) &= -\mathrm{KL}(q_{\phi_c}(c|x)\|p(C)) - \mathrm{KL}(q_{\phi_s}(s|x)\|p(S)) \\
&\quad + \mathbb{E}_{c\sim q_{\phi_c}(c|x)} \left[ \log q_{\theta_c}(y|c) \right] + \mathbb{E}_{s\sim q_{\phi_s}(s|x)} \left[ \log q_{\theta_s}(d|s) \right] \\
&\quad + \mathbb{E}_{c\sim q_{\phi_c}(c|x),s\sim q_{\phi_s}(s|x)} \left[ \log p_{\theta_x}(x|c, s) \right],
\end{aligned}
\tag{10}
$$

which is the final ELBO in the main paper (Eq. (4)).

## C    COMPLEMENTARY EXPERIMENTAL DETAILS

In this section, we provide complementary experimental details. Appendix C.1 contains introduction of all datasets. In Appendix C.2, we present more implementation details, including baselines, models, training details, evaluation metrics, and running environments. Appendix C.3 contains network architectures of our H-NTL. In Appendix C.4, we provide details about the style augmentations in our source-only H-NTL. Appendix C.5 contains the full training algorithm of the source-only H-NTL. Finally, in Appendix C.6, we provide a detailed process for conducting the ablation study.

### C.1    DATASETS

Our experiments involve three basic tasks on digit datasets and three challenging tasks on real-world datasets. The digit tasks contain three random-selected pairs from four digit datasets: *MNIST* (MT) (Deng, 2012), *MNIST-M* (MM) (Ganin et al., 2016), *SVHN* (SN) (Netzer et al., 2011) and *SYN-D* (SD) (Roy et al., 2018). For challenging tasks, we involve *CIFAR10* to *STL10* (Coates et al., 2011) (C10→S10), *VisDA* (Peng et al., 2017) (VT→VV) and *OfficeHome* (Venkateswara et al., 2017) (OP→OC). We provide more detailed introductions for each dataset as follows:

- ***Digits:*** Following Wang et al. (2022b), we involve digit datasets for evaluation. Digit datasets include MNIST (MT) (Deng, 2012), USPS (US) (Hull, 1994), SVHN (SN) (Netzer et al., 2011), MNIST-M (MM) (Ganin et al., 2016) and SYN-D (SD) (Roy et al., 2018). Each dataset contains ten digits collected from real scenes or artificially constructed. We conduct experiments on three random-selected pairs: MM→MT, SN→SD, and SD→MT.
- ***CIFAR10 & STL10:*** We also follow Wang et al. (2022b) to evaluate on CIFAR10 (C10) and STL10 (S10) (Coates et al., 2011) . Both C10 and S10 are ten-class classification datasets, which contain six animal categories and four transportation categories. We use C10 as the source domain and S10 as the target domain.
- ***VisDA:*** VisDA (Peng et al., 2017) contains a training set VisDA-T (VT) and a validation set VisDA-V (VV) of 12 object categories. Following Wang et al. (2022b), we consider the non-transferable task from VT to VV.
- ***OfficeHome:*** OfficeHome (Venkateswara et al., 2017) contains four domains, where each domain consists of 65 categories. The number of categories in OfficeHome is significantly more than the above datasets, and thus, it's more challenging for NTL methods to resist transferability. We conduct NTL experiments from Product (OP) to Clipart (OC), where OP contains product images without backgrounds and OC is a collection of clipart images.

### C.2    IMPLEMENTATION DETAILS

**Baselines.** We use supervised learning (SL) and the non-transferable learning method proposed by Wang et al. (2022b) as baselines. In order to avoid confusion between the method name of Wang et al. (2022b) and the task name, we denote their methods as tNTL and sNTL for the target-specified and source-only tasks, respectively. We provide brief introductions for baseline methods as follows:

- **SL:** We use a standard supervised learning pipeline with cross-entropy loss.
- **tNTL:** a target-specified NTL method inspired by information bottleneck. This method adds two statistical dependence relaxation terms on standard supervised learning to resist transferability: (i) maximizing the Kullback-Leible (KL) divergence between target domain representation and label, and (ii) maximizing the maximum mean discrepancy (MMD) between the distribution of source and target domain representations.
- **sNTL:** a source-only NTL method, where a generative adversarial network (GAN) is used to synthesize fake target domain data with domain shifts. Thus, the source-only NTL can be solved by the tNTL method.

**Models.** For a fair comparison, we follow the backbones in (Wang et al., 2022b) to conduct all experiments. Specifically, for the backbones in VAE encoder modules ($\hat{q}_{\phi_c}$ and $\hat{q}_{\phi_s}$) and student network $f_{ntl}$, we apply VGG-11 (Simonyan & Zisserman, 2015) in digit tasks (MM→MT, SN→SD and SD→MT), VGG-13 (Simonyan & Zisserman, 2015) in C10→S10, and VGG-19 in VT→VV and OP→OC. All backbone networks are initialized as the pre-trained version of ImageNet-1K (Deng et al., 2009). The detailed network architectures of the VAE encoder modules, the VAE classifier modules, the VAE decoder modules, and the student network are shown in Appendix C.3.

**Training details.** For basic digit tasks, we resize images to 32×32. For other challenging tasks, we resize images to two different resolutions: 32×32 and 64×64. For training SL, we employ the SGD as an optimizer with $lr = 0.001$ and set the batch size to 32. For tNTL and sNTL, we use their released code[7] and the same hyperparameters settings reported in their paper to run experiments. For our proposed H-NTL, we employ the SGD as an optimizer with $lr = 0.1$ and set the batch size to 128. The disentanglement VAE is trained for 20 epochs, and the dual-path knowledge distillation is trained for 30 epochs. The hyper-parameter $\lambda_t$ is set to 1.0 for all datasets. In addition, the number of image style augmentations is set to 10 when performing source-only NTL tasks. Following Wang et al. (2022b), we randomly select 8,000 samples as training data and 1,000 samples as testing data without overlap for digit tasks, C10→S10, and VT→VT. For OP→OC, we use 3,000 for training data and 1,000 samples due to the limitation of the dataset size.

**Evaluation metric.** We show top-1 classification accuracy ($Acc$) on source/target domain respectively. Results are reported as average with standard deviation over three independent runs. In addition, we calculate the accuracy drop ($\Delta = Acc_{sl} - Acc_{ntl}$) and relative drop ($\Delta\% = \Delta/Acc_{sl}$) on source/target domain, respectively.

**Environment.** Our code is implemented in Python 3.8.8 and PyTorch 1.8.0. All experiments are conducted on a server running Ubuntu 20.04 LTS, equipped with an NVIDIA RTX A6000 GPU.

## C.3 NETWORK ARCHITECTURE

In this section, we present the detailed network architectures of our VAE encoder modules ($\hat{q}_{\phi_c}$ and $\hat{q}_{\phi_s}$), VAE classifier modules ($\hat{p}_{\theta_c}$ and $\hat{p}_{\theta_s}$), VAE decoder module ($\hat{p}_{\theta_x}$) and the student network $f_{ntl}$. In the VAE encoder modules ($\hat{q}_{\phi_c}$ and $\hat{q}_{\phi_s}$) and the student network $f_{ntl}$, we follow Wang et al. (2022b) to use several popular architectures as backbones to extract image features. Specifically, we apply VGG-11 (Simonyan & Zisserman, 2015) in digit tasks (MM→MT, SN→SD and SD→MT), VGG-13 in C10→S10, and VGG-19 in VT→VV and OP→OC.

First of all, the architecture of the VAE encoder modules ($\hat{q}_{\phi_c}$ and $\hat{q}_{\phi_s}$) is shown in Table 5. Specifically, **img** is a parameter corresponding to the input image size. If the image size = 32×32 (for the three basic digit tasks and three challenging tasks), the **img** = 1. If the image size = 64×64 (only for the three challenging tasks), the **img** = 2.

Table 5: The architecture of the VAE encoder modules ($\hat{q}_{\phi_c}$ and $\hat{q}_{\phi_s}$), where the parameter **img** corresponds to the input size.

| **Latent Mapping** | Mean $\mu$ | Variance $\sigma$ |
|---|---|---|
| | Linear(512, 256) | Linear(512, 256) |
| **Pooling Layer** | AdaptiveAvgPooling([512, **img**, **img**], 512) | |
| **Feature Extractor** | Backbone Network (VGG-11/VGG-13/VGG-19) | |

Then, the architecture of the VAE classifier modules ($\hat{p}_{\theta_c}$ and $\hat{p}_{\theta_s}$) is shown in Table 6. In particular, the *content factor* classifier $\hat{p}_{\theta_c}$ contains a linear layer which maps the estimated 256-dim *content factor* $C$ to the classification space (10-dim for digits and C10→S10, 12-dim for VV→VT, and 65-dim for OP→OC). Similarly, through a linear layer, the *style factor* classifier $\hat{p}_{\theta_s}$ maps the estimated 256-dim *style factor* $S$ to the domain classification space whose dimension equals to domain_num (in this paper, the domain_num = 2).

Table 6: The architecture of the VAE classifier modules ($\hat{p}_{\theta_c}$ and $\hat{p}_{\theta_s}$). The class_num equals to the number of categories, and the domain_num equals to the number of domains (= 2 in this paper).

| **Classifier** | *content factor* | *style factor* |
|---|---|---|
| | Linear(256, class_num) | Linear(256, domain_num) |

Next, the architecture of the VAE decoder modules (i.e., $\hat{p}_{\theta_x}$) is shown in Table 7. Particularly, the **Conv4** layer only exists when the input image size equals to $64 \times 64$. The input layer of the decoder

---

[7] https://github.com/conditionWang/NTL

Table 7: The architecture of the VAE decoder module ($\hat{p}_{\theta_x}$). The **Conv4** layer will be removed if the input image size equals to 32×32.

| | |
|---|---|
| **Output** | Sigmoid, Conv2d(32, 3, 1, 1) |
| | LeakyReLU, BatchNorm(32) |
| | ConvTranspose2d(32, 32, 3, 2, 1, 1) |
| **Conv4** | LeakyReLU, BatchNorm(32) |
| | ConvTranspose2d(32, 32, 3, 2, 1, 1) |
| **Conv3** | LeakyReLU, BatchNorm(32) |
| | ConvTranspose2d(64, 32, 3, 2, 1, 1) |
| **Conv2** | LeakyReLU, BatchNorm(64) |
| | ConvTranspose2d(128, 64, 3, 2, 1, 1) |
| **Conv1** | LeakyReLU, BatchNorm(128) |
| | ConvTranspose2d(256, 128, 3, 2, 1, 1) |
| **Input** | Linear(512, 1024) |
| | Concat |
| | *content factor* \| *style factor* |

first concat the 256-dim *content factor* $C$ and the *style factor* $S$, and then maps them to a high dimension feature to perform transposed convolution. The parameters of the transposed convolution operation (i.e., ConvTranspose2d[8]) are: in_channels, out_channels, kernel_size, stride, padding, and output_padding. In the final output layer, the parameters of the convolution operation (i.e., Conv2d[9]) are: in_channels, out_channels, kernel_size, stride, and padding.

Finally, the architecture of the student model $f_{ntl}$ is shown in Table 8. The architecture is the same as Wang et al. (2022b) for a fair comparison. Especially, if the input image size equals to 32×32, the parameter **img** = 1. If the image size equals to 64×64 (only for the three challenging tasks), the **img** = 1. Besides, the class_num corresponds to the number of categories in each dataset.

Table 8: The architecture of student models $f_{ntl}$, where the parameter **img** corresponds to the input size. The class_num equals to the number of categories.

| | |
|---|---|
| **Classifier** | Linear(256, class_num). |
| | Linear(256, 256), ReLU, Dropout; |
| | Linear(512\***img**\***img**, 256), ReLU, Dropout; |
| **Feature Extractor** | Backbone Network (VGG-11/VGG-13/VGG-19). |

## C.4 STYLE AUGMENTATION FOR SOURCE-ONLY NTL TASK

In this section, we provide details about the style augmentations used in our source-only H-NTL. As mentioned in the main paper, our aim is to impose interventions on the *style factor* $S$ (i.e., $do(S)$), which do not impact $P(Y|C)$. In practice, we cannot obtain naturally entangled *content factor* $C$ and *style factor* $S$ from a single source domain and subsequently manipulate the *style factor* value. Thus, we leverage image style augmentations (Sohn et al., 2020; Berthelot et al., 2019; Saito et al., 2021; Zheng et al., 2024) to implicitly impose interventions on source domain data. Our image style augmentations are derived from RandAugment (Cubuk et al., 2020). For completeness, we briefly present the list of image style augmentations we used in Table 9. For more details and corresponding parameters, please refer to Cubuk et al. (2020). Specifically, in our experiments, all hyper-parameters in these style augmentations follow Saito et al. (2021), and we do not tune any hyper-parameters.

In Fig. 6 and Fig. 7, we show the original and augmented source domain data on C10 and VT, respectively. Each source domain image is augmented by a kind of style augmentation randomly selected from the Table 9. We can see that these style augmentations do not influence the content factor but significantly change the image style, thus satisfying our intervention aims.

---

[8]https://pytorch.org/docs/1.8.0/generated/torch.nn.ConvTranspose2d.html

[9]https://pytorch.org/docs/1.8.0/generated/torch.nn.Conv2d.html

Table 9: List of image style augmentations. We only present names with brief descriptions. For details and corresponding parameters, please refer to (Cubuk et al., 2020).

| Augmentation | Brief Description |
|---|---|
| Autocontrast | Maximizes the image contrast by setting the darkest (lightest) pixel to black (white), and then blends with the original image. |
| Brightness | Adjusts the brightness of the image. |
| Color | Adjusts the color balance of the image. |
| Contrast | Controls the contrast of the image. |
| Cutout | Sets a random square patch of side-length pixels to gray. |
| Equalize | Equalizes the image histogram, and then blends with the original image. |
| Invert | Inverts the pixels of the image, and then blends with the original image. |
| Posterize | Reduces each pixel to a certain bit. |
| Rotate | Rotates the image. |
| Sharpness | Adjusts the sharpness of the image. |
| Shear_x | Shears the image along the horizontal axis. |
| Shear_y | Shears the image along the vertical axis. |
| Solarize | Inverts all pixels above a certain threshold value. |
| Translate_x | Translates the image horizontally by certain pixels. |
| Translate_y | Translates the image vertically by certain pixels. |

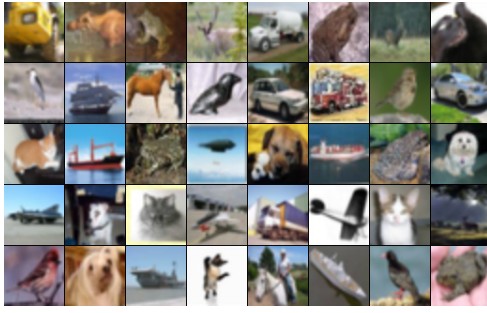
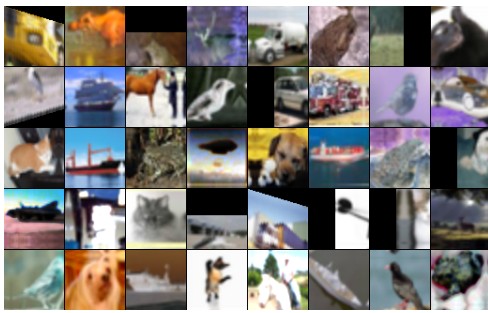

(a) Original images                    (b) Augmented images

Figure 6: Comparison of original images and augmented images on C10.

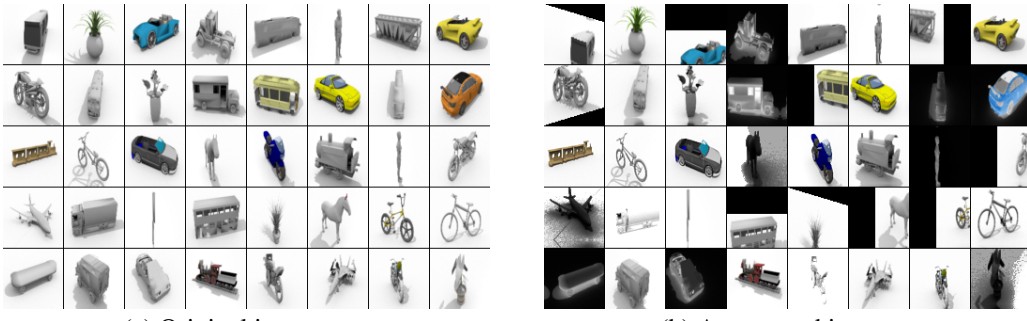

(a) Original images                    (b) Augmented images

Figure 7: Comparison of original images and augmented images on VT.

## C.5 FULL ALGORITHM OF SOURCE-ONLY H-NTL

In this section, we provide the full algorithm for training source-only H-NTL. Briefly, in the source-only NTL task, we conduct image style augmentations on source domain data to obtain OOD data. Then, we see all OOD data augmented by different styles as the target domain, and we use the accessible source domain and the target domain to train the H-NTL in the same way as target-specified NTL. The detailed training procedure of source-only H-NTL is shown in Algorithm 2.

Specifically, the augmentation pool $\mathcal{A} = \{a_i\}_{i=1}^{N_a}$ contains $N_a$ kinds of style augmentations which are randomly selected from the total style augmentation list (see Appendix C.4 and Table 9), where $N_a$ is a hyper-parameter. In both disentanglement and dual-path knowledge distillation, we randomly select different style augmentations from the pool $\mathcal{A}$ to augment each image in the source domain $\mathcal{D}_s$. This augmentation process is repeated in every epoch (i.e., the inner for-loop in the disentanglement and dual-path knowledge distillation), effectively expanding the distribution range of the augmented source domain data.

---

**Algorithm 2** Train source-only H-NTL

---

1: Training set in source domain $\mathcal{D}_s = \{x_i, y_i, d_i = 0\}_{i=1}^{N_s}$; Augmentation pool $\mathcal{A} = \{a_i\}_{i=1}^{N_a}$; Batch-size $n$; Disentanglement epoch $E_{dis}$, KD epoch $E_{kd}$, VAE modules $\{\hat{q}_{\phi_c}, \hat{q}_{\phi_s}, \hat{p}_{\theta_x}, \hat{p}_{\theta_y}, \hat{p}_{\theta_d}\}$ and the NTL network $f_{ntl}$.
2: **for** $i = 1$ to $E_{dis}$ **do**
3:     **for** $j = 1$ to $N_s$ **do**
4:         Fetch the $j$-th sample $x_j$ in the source domain $\mathcal{D}_s$;
5:         Random select augmentation function $a \in \mathcal{A}$;
6:         Perform image style augmentation on the $j$-th sample: $\hat{x}_j = a(x_j)$;
7:     **end for**
8:     See augmented data as target domain $\mathcal{D}_t = \hat{\mathcal{D}}_s = \{\hat{x}_j, y_j, d_j = 1\}_{j=1}^{N_s}$;
9:     Using $\mathcal{D}_s \cup \mathcal{D}_t$ to train VAE $\{\hat{q}_{\phi_c}, \hat{q}_{\phi_s}, \hat{p}_{\theta_x}, \hat{p}_{\theta_y}, \hat{p}_{\theta_d}\}$ by maximizing ELBO;
10: **end for**
11: **for** $i = 1$ to $E_{kd}$ **do**
12:     **for** $j = 1$ to $N_s$ **do**
13:         Fetch the $j$-th sample $x_j$ in the source domain $\mathcal{D}_s$;
14:         Random select augmentation function $a \in \mathcal{A}$;
15:         Performing image style augmentation on the $j$-th sample: $\hat{x}_j = a(x_j)$;
16:     **end for**
17:     See augmented data as target domain $\mathcal{D}_t = \hat{\mathcal{D}}_s = \{\hat{x}_j, y_j, d_j = 1\}_{j=1}^{N_s}$;
18:     Using $\mathcal{D}_s \cup \mathcal{D}_t$ to train $f_{ntl}$ through minimizing $\mathcal{L}_{kd}$;
19: **end for**

---

### C.6 THE DETAILED PROCESS FOR CONDUCTING THE ABLATION STUDY

The ablation studies in the main paper aim to explore the effectiveness of the main components (the *content factor $C$* and the *style factor $S$*) in the H-NTL. To validate the importance of $C$ for H-NTL, we train the student model to only fit $C$ of the source domain by using a single-path knowledge distillation (denoted as H-NTL w/o $C$). The only difference between H-NTL w/o and the original H-NTL is that the target domain path is released. Similarly, to validate the importance of $S$, we train the student model to only fit $S$ of the target domain (denoted as H-NTL w/o $S$). Then, we compare them with the original H-NTL in our paper, i.e., H-NTL (full). The disentanglement processes are the same for H-NTL w/o $C$, H-NTL w/o $S$, and H-NTL (full).

## D ADDITIONAL EXPERIMENTS AND ANALYSES

This section contains additional experiments and analyses. In Appendix D.1, we show results of the proposed H-NTL on high-resolution images. In Appendix D.2, we show results of H-NTL on intellectual property protections. In Appendix D.3, we compare the proposed H-NTL with an additional baseline. In Appendix D.4 and Appendix D.5, we analyze the influence of main hyperparameters in our H-NTL. In Appendix D.6, we empirically analyze the disentanglement between *content factors* and *style factors*. In Appendix D.7, we present visualization results.

### D.1 RESULTS ON HIGH RESOLUTION DATASETS

We run additional evaluations with higher-resolution images. In detail, we run experiments on the original resolution of VT→VV (112×112) (Peng et al., 2017) and OP→OC (224×224) (Venkateswara et al., 2017) without resizing, thus showing the effectiveness of the proposed H-NTL on higher

resolution images. Results of the proposed H-NTL on the target-specified NTL setting and source-only NTL setting are shown in Table 10 and Table 11, respectively. On higher-resolution datasets, the proposed H-NTL can still effectively maintain source domain performance (comparable to SL) and significantly degrade the target domain performance.

Table 10: Results of **target-specified** H-NTL on high-resolution datasets.

| Source domain | Target domain | Image resolution | SL (source/target) | H-NTL (source/target) | source drop | target drop |
|---|---|---|---|---|---|---|
| VT | VV | 32×32 | 89.7/22.4 | 91.7/8.1 | -2.0 | 14.3 |
| VT | VV | 64×64 | 93.4/35.6 | 94.6/8.2 | -1.2 | 27.4 |
| VT | VV | **112×112** | 97.5/42.6 | 97.8/8.3 | -0.3 | 34.3 |
| OP | OC | 32×32 | 65.6/23.6 | 65.6/5.5 | 0.0 | 18.1 |
| OP | OC | 64×64 | 75.6/31.6 | 76.4/6.7 | -0.8 | 24.9 |
| OP | OC | **224×224** | 86.0/35.5 | 84.9/3.6 | 1.1 | 31.9 |

Table 11: Results of **source-only** H-NTL on high-resolution datasets.

| Source domain | Target domain | Image resolution | SL (source/target) | H-NTL (source/target) | source drop | target drop |
|---|---|---|---|---|---|---|
| VT | VV | 32×32 | 89.7/22.4 | 91.6/9.3 | -1.9 | 13.1 |
| VT | VV | 64×64 | 93.4/35.6 | 94.9/7.8 | -1.5 | 27.8 |
| VT | VV | **112×112** | 97.5/42.6 | 96.7/8.2 | 0.8 | 34.4 |
| OP | OC | 32×32 | 65.6/23.6 | 63.7/14.4 | 1.9 | 9.2 |
| OP | OC | 64×64 | 75.6/31.6 | 71.6/16.0 | 4.0 | 15.6 |
| OP | OC | **224×224** | 86.0/35.5 | 83.3/10.1 | 2.7 | 25.4 |

## D.2 RESULTS ON IP PROTECTIONS

The practical application of the proposed H-NTL lies in the model intellectual property (IP) protections. Our H-NTL can provide two kinds of IP protection: ownership verification, and applicability authorization. As you suggested, we discuss and run more experiments on ownership verification and applicability authorization to further show the effectiveness of our H-NTL on real-world applications.

**Ownership verification.** The aim of ownership verification is to verify the ownership of a deep learning model. The proposed H-NTL provides a solution for ownership verification by triggering misclassification on the target domain (Wang et al., 2022b). Specifically, we can add a pre-defined trigger patch (only known by the model owner) on the original dataset (i.e., the source domain) and see them as the target domain. It is worth noting that such a pre-defined trigger patch can be controlled to be shallow so that normal supervised learning (SL) models trained on the original source domain can still have normal performance on the patched source domain. Then, we train a deep learning model by using the proposed H-NTL on these two domains. After that, the training model will perform poorly on the data with the patch but have a good performance on the data without the patch. Thus, by observing the performance difference of a trained model on the source domain data with and without the pre-defined trigger patch, we can verify whether a deep learning model belongs to the model owner.

We have verify the strong performance of the proposed H-NTL on watermark data with different patch values (results are shown in Section 4.1 and Fig. 4 in the main paper), which serves as the core task of ownership verification. Here we run additional experiments of ownership verification, and results are shown in Table 12. We can see that models trained with H-NTL behave differently on the data with and without the patch, whereas SL models perform nearly the same. Accordingly, the ownership of trained models can be verified.

**Applicability authorization.** Applicability authorization aims at authorizing models to certain data for preventing their usage on unauthorized data (Wang et al., 2022b), which can be solved by applying source-only H-NTL to restrict the model generalization ability to only the authorized domain. Specifically, we add a pre-defined authorized patch to the original data to be authorized and see them as the source domain. We regard *the union of the original data (without the authorized*

Table 12: Results of H-NTL on ownership verification. "(p)" means patched data. The results of H-NTL are highlighted in **bold**.

| Source domain | Target domain | SL (patch/non-patch) | H-NTL (patch/non-patch) |
|---|---|---|---|
| MT | MT(p) | 98.8/98.5 | **98.6/10.6** |
| US | US(p) | 98.9/98.8 | **99.4/14.3** |
| SN | SN(p) | 87.9/87.4 | **87.8/8.5** |
| MM | MM(p) | 92.8/92.0 | **93.4/9.7** |
| SD | SD(p) | 96.3/95.9 | **96.1/9.5** |
| C10 | C10(p) | 86.5/60.2 | **87.9/8.9** |
| S10 | S10(p) | 88.1/82.0 | **89.5/10.9** |
| VT | VT(p) | 93.4/92.2 | **95.1/8.3** |

*patch), the augmented original data with and without the authorized patch* as the target domain. Then, we train a deep learning model by using the proposed H-NTL on these two domains. After that, the trained model will only perform well on the authorized data (i.e., the original data with the authorized patch). For unauthorized data (e.g., the original data without the authorized patch, the data from other domains with or without the authorized patch), the trained model will perform poorly. Thus, we achieve the model applicability authorization.

We run experiments of applicability authorization, and results are shown in Table 13 (Digits datasets) and Table 14 (complex datasets). We can see that the model trained by H-NTL has good performance on the authorized domain (highlighted in **bold** in each row) and poor performance on all unauthorized domains. This shows the effectiveness of H-NTL in applicability authorization.

Table 13: Results of H-NTL on applicability authorization (digits datasets). "(p)" means data with the authorized patch. In each row, the authorized domain is highlighted in **bold**. The last column shows the averaged results on authorized and unauthorized domains (also highlighted in **bold**)

| Source domain | MT (p) | US (p) | SN (p) | MM (p) | SD (p) | MT | US | SN | MM | SD | H-NTL (auth/unauth) |
|---|---|---|---|---|---|---|---|---|---|---|---|
| MT(p) | **98.2** | 12.3 | 10.1 | 12.3 | 9.6 | 9.2 | 9.7 | 10.1 | 8.4 | 9.6 | **98.2/10.1** |
| US(p) | 9.5 | **99.2** | 6.0 | 10.8 | 10.2 | 9.5 | 17.2 | 6.0 | 10.8 | 10.2 | **99.2/8.9** |
| SN(p) | 16.5 | 18.8 | **88.3** | 11.9 | 10.5 | 8.9 | 9.2 | 17.0 | 10.3 | 10.0 | **88.3/12.6** |
| MM(p) | 20.3 | 17.2 | 14.5 | **92.2** | 10.9 | 9.7 | 17.2 | 5.8 | 12.5 | 11.0 | **92.2/13.2** |
| SD(p) | 8.7 | 29.5 | 22.0 | 17.1 | **94.4** | 8.1 | 7.5 | 9.3 | 10.9 | 11.9 | **94.4/13.9** |

Table 14: Results of H-NTL on applicability authorization (complex datasets). "(p)" means data with the authorized patch. In each row, the authorized domain is highlighted in **bold**. The last column shows the averaged results on authorized and unauthorized domains (also highlighted in **bold**).

| Source domain | C10(p) | S10(p) | C10 | S10 | H-NTL(auth/unauth) |
|---|---|---|---|---|---|
| C10(p) | **87.4** | 9.7 | 10.6 | 8.9 | **87.4/9.7** |
| S10(p) | 12.7 | **88.3** | 8.9 | 11.2 | **88.3/10.9** |
| Source domain | VT(p) | VV(p) | VT | VV | H-NTL(auth/unauth) |
| VT(p) | **92.0** | 8.6 | 10.9 | 9.5 | **92.0/9.7** |

## D.3 MORE BASELINES

For a comprehensive evaluation, we further design an additional baseline method (denoted as TargetClass) which can be used for non-transferable learning. Specifically, considering an N-class non-transferable learning task, we take the same backbone as our main paper as a feature extractor and use a classification head with N+1 classes to predict the N classes in the source domain, but predict an additional "target class" as class N+1 for all samples in the target domain. Therefore, the model trained by TargetClass will map all the target domain data to an isolated cluster that stays away from the N source-domain-class clusters in the feature space, thus hindering the source-to-target knowledge transfer.

We run experiments under the same setting as our main paper, and results are shown in Table 15 (on natural datasets) and Table 16 (on watermark target-domain). From the comparison, the proposed H-NTL generally has better performance.

Table 15: Comparison of the new baseline (denoted as TargetClass) and the proposed H-NTL on natural data. The best results are highlighted in **bold**.

| Source domain | Target domain | Image resolution | TargetClass (source/target) | H-NTL (source/target) |
|---|---|---|---|---|
| MM | MT | 32×32 | **92.8/9.5** | 93.1/9.9 |
| SN | SD | 32×32 | 87.8/10.4 | **88.1/9.2** |
| SD | MT | 32×32 | 97.1/12.6 | **97.1/11.0** |
| C10 | S10 | 32×32 | 81.3/50.2 | **80.6/28.1** |
| C10 | S10 | 64×64 | 85.8/22.0 | **87.6/9.6** |
| VV | VT | 32×32 | 91.4/10.9 | **91.7/8.1** |
| VV | VT | 64×64 | 94.2/7.9 | **94.6/8.2** |
| OP | OC | 32×32 | 64.2/8.6 | **65.6/5.5** |
| OP | OC | 64×64 | 72.2/6.6 | **76.4/6.7** |

Table 16: Comparison of the new baseline (denoted as TargetClass) and the proposed H-NTL on natural data. The best results are highlighted in **bold**.

| Source domain | Target domain | Patch value | TargetClass (source/target) | H-NTL (source/target) |
|---|---|---|---|---|
| C10 | C10(p) | 20 | 83.0/10.7 | **87.8/7.9** |
| C10 | C10(p) | 40 | 85.5/10.0 | **88.7/10.1** |
| C10 | C10(p) | 60 | 85.5/8.8 | **87.3/9.8** |
| S10 | S10(p) | 20 | 85.4/53.4 | **83.6/16.7** |
| S10 | S10(p) | 40 | 85.9/20.4 | **87.4/12.5** |
| S10 | S10(p) | 60 | 86.9/15.9 | **88.3/9.0** |

## D.4 INFLUENCE OF KD WEIGHT

The main hyper-parameter in our proposed H-NTL is the KD weight $\lambda_t$ which is used to balance the loss values from source and target domain distillation paths. Actually, we always set the weight $\lambda_t$ to 1 and do not need to tune its value. In order to analyze its influence on the total NTL performance, we change its value in a large range (from $10^{-4}$ to 10) and conduct both target-specified/source-only NTL tasks on C10→S10 and VT→VV. The results of target-specified and source-only NTL tasks are shown in Fig. 8 and Fig. 9, respectively. In the target-specified NTL task, we can see that if we set the $\lambda_t$ too small, the network $f_{ntl}$ is difficult to fit the *style factors* in the target domain, and thus, the target domain performance cannot be degraded. If the weight $\lambda_t$ is set too large, the network $f_{ntl}$ will be hard to fit the *content factors* in the source domain, which leads to the failure of source domain performance maintenance. In the source-only NTL task, the influences of $\lambda_t$ on maintaining source domain performance and degrading target domain performance are both similar to the target-specified NTL task. One difference is that the total NTL performance of source-only H-NTL is more vulnerable to the weight $\lambda_t$ due to the unseen target domain in source-only setting.

## D.5 INFLUENCE OF THE NUMBER OF STYLE AUGMENTATION IN SOURCE-ONLY TASKS

A main hyper-parameter in source-only H-NTL is the number of image style augmentations. To analyze its influence on the total performance of source-only NTL, we change its value from 1 to 12. The results on C10→S10 and VT→VV are shown in Fig. 10. We can see that the source domain performance is not sensitive to the number of augmentations, but the target domain performance degradation is limited under a small number of augmentations (e.g., ≤3). This limitation arises from the restricted distribution of the augmented source domain data which is augmented with a small number of styles and fails to encompass a sufficient range of out-of-distribution data. Thus, the target domain performance degradation is hard to generalize to the real unseen target domain.

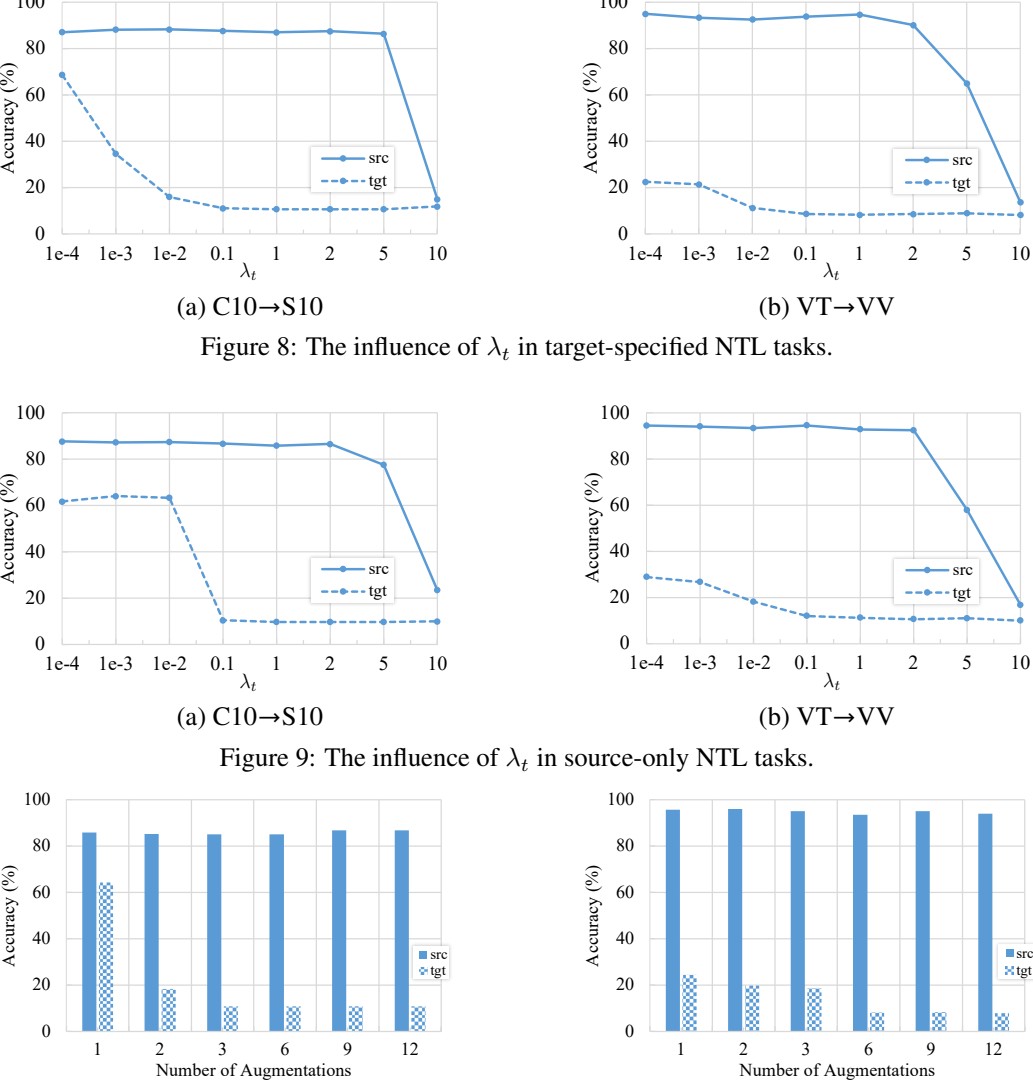

Figure 8: The influence of $\lambda_t$ in target-specified NTL tasks.

Figure 9: The influence of $\lambda_t$ in source-only NTL tasks.

Figure 10: The influence of the number of style augmentations in source-only tasks.

## D.6 DISENTANGLEMENT OF CONTENT AND STYLE FACTORS

In this section, we empirically analyze the effectiveness of the disentanglement between *content factors* and *style factors*. For experimental validation, we conduct cross-prediction. In detail, we employ the VAE encoders ($\hat{q}_{\phi_c}$ and $\hat{q}_{\phi_s}$) to extract the *content factor* $C$ and *style factor* $S$ from data, and then we use both of them to predict *class* $Y$ and *domain* $D$ through VAE classifiers ($\hat{p}_{\theta_y}$ and $\hat{p}_{\theta_d}$). The results are shown in Fig. 11, with subfigures (a) to (d) presenting the predictions of $C$-to-$Y$, $C$-to-$D$, $S$-to-$Y$, and $S$-to-$D$, respectively.

The results show that only *content factors* are meaningful in predicting *class*, and *style factors* are meaningful in predicting *domain*. If we use *content factors* to predict *domain* (or use *style factors* to predict *class*), the results are similar to random classification. These phenomenons empirically indicate the effectiveness of our disentanglement between *content factors* and *style factors*.

## D.7 VISUALIZATION

**t-SNE visualization of non-transferable feature representations.** As shown in Fig. 12 and Fig. 13, we use t-SNE visualization (Van der Maaten & Hinton, 2008) to present the learned non-transferable representations on C10→S10 and VT→VV, respectively. For intuitive comparison, we present feature

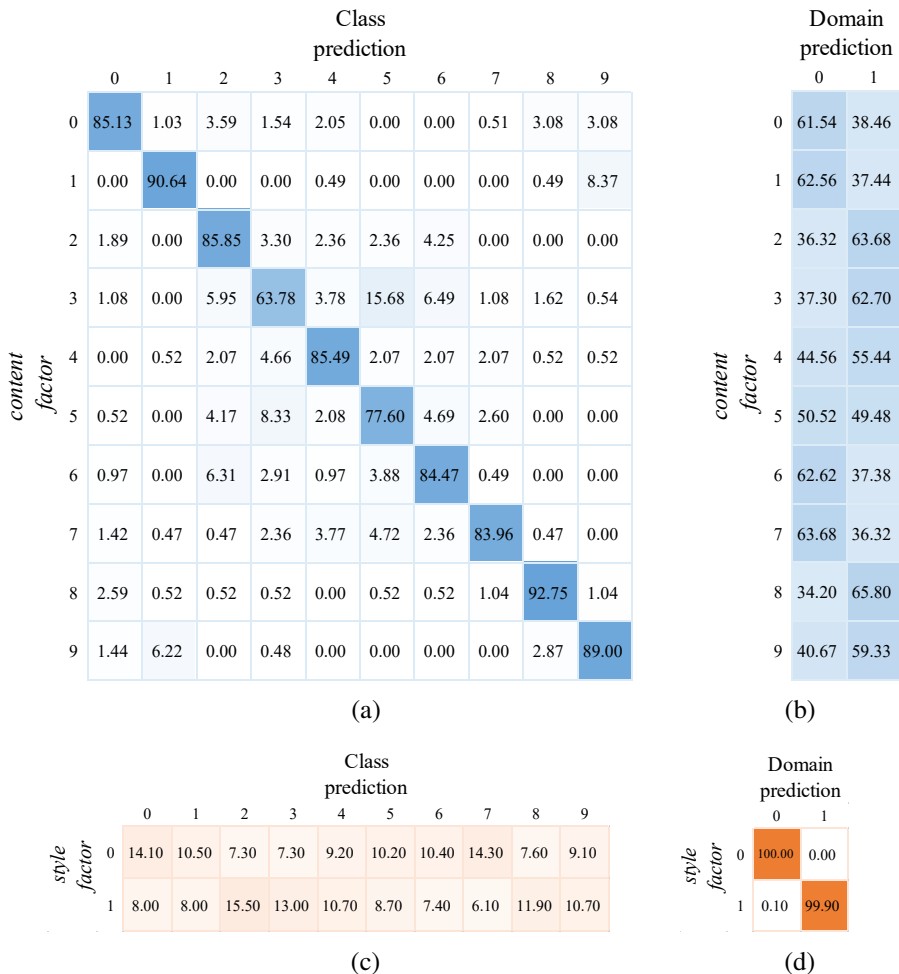

Figure 11: Visualization of disentanglement between the *content factor* and the *style factor* on C10→S10. The number in each cell denotes the top-1 classification accuracy (%).

representations learned by supervised learning (SL), target-specified H-NTL, and source-only H-NTL. Particularly, those opaque dots represent source domain features, and transparent dots represent target domain features. As shown in Fig. 12 (a) and Fig. 13 (a), although only trained in the source domain, the target domain features of SL are overlapped with the source domain features and maintain a certain discriminability. This leads to the meaningful performance of SL on the target domain. Compared to SL, both target-specified and source-only H-NTL can clearly separate the source and target domain features with a certain distance. In addition, the feature representations in the target domain are randomly distributed due to the constraint of fitting *style factors* in the target domain distillation path. Thus, the target domain performance is degraded to the accuracy of random classification. Moreover, because of the disentanglement from paired source-target domains, the discriminability of features in the source domain is maintained (or even enhanced) compared to SL.

# E   MORE DISCUSSION OF THE CAUSAL MODEL

In this section, we provide further explanation and discussion of our causal model. In Appendix E.1, we discuss the statistical dependence between the *content factor* $C$ and the *style factor* $S$ in our causal model. In Appendix E.2, we provide more explanation about the causal direction from the *style factor* $S$ to the *domain* $D$. In addition, we also consider another causal direction (i.e., $D$ causes $S$) and conduct basic experiments.

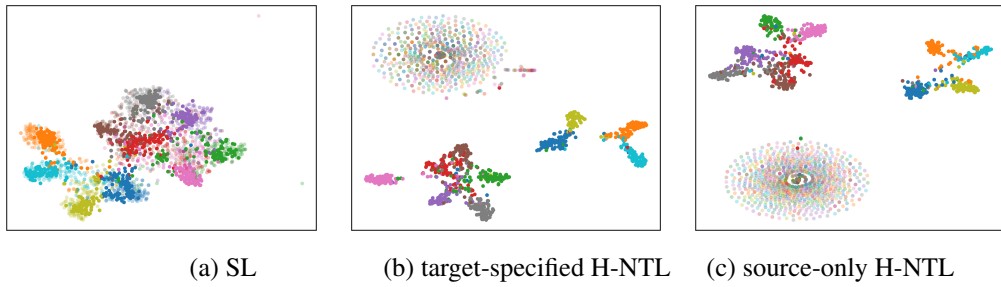

(a) SL  (b) target-specified H-NTL  (c) source-only H-NTL

Figure 12: t-SNE Visualization of features on C10→S10. Different color denotes different class, and opaque/transparent dots represent source/target domain data.

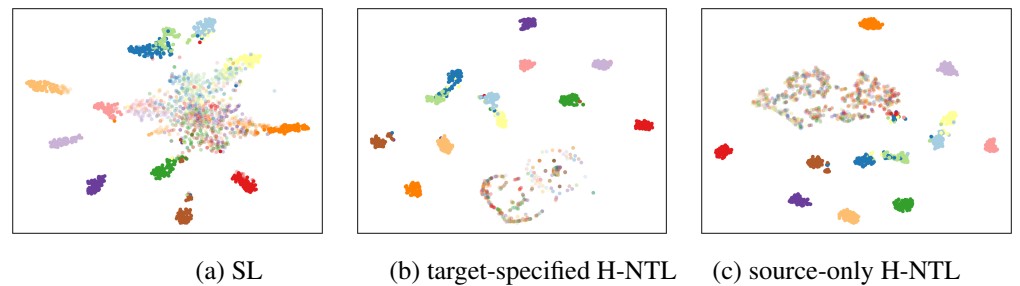

(a) SL  (b) target-specified H-NTL  (c) source-only H-NTL

Figure 13: t-SNE Visualization of features on VT→VV. Different color denotes different class, and opaque/transparent dots represent source/target domain data.

### E.1 DEPENDENCE OF THE CONTENT FACTOR $C$ ON THE STYLE FACTOR $S$

**The definition of the content factor $C$ and the style factor $S$.** In order to avoid being misled to fit spurious correlation and fake independence, we distinguish the unobservable and naturally mixed *contents* and *styles* by separately modeling them in our causal model. As shown in Fig. 2 in the main paper, we decompose an instance $X$ into two latent variables:

- **Content factor $C$**: This corresponds to the intrinsic, class-related information that is a cause of the label $Y$ in both the source and the target domain. For example, in the case where we use CIFAR10 as the source domain and STL10 as the target domain, $C$ represents the object category that is common to both the CIFAR10 and STL10 (e.g., bird, airplane, deer, etc.).
- **Style factor $S$**: Except for the content factor $C$, all other factors can belong to style factors. For example, in the NTL task from the wildlife park to the zoo (Fig. 1 in the main paper), the environments of wild and zoo are *style factors*. In the toy experiment (Appendix A), both the digit colors and the digit typefaces belong to *style factors*.

**The motivation of modeling the statistical dependence between $C$ and $S$.** As mentioned in the introduction (Section 1 in the main paper), *the common co-occurrence between contents and styles in real-world scenarios* causes that existing methods inadvertently fitting (i) *spurious correlation between styles and labels*, and (ii) *fake independence between contents and labels*. We assume the co-occurrence between *contents* and *styles* is caused by latent confounders (Von Kügelgen et al., 2021; Schölkopf, 2022; Klindt et al., 2021), and thus, we model it by a statistical dependence between the *style factor $S$* and the *content factor $C$* in our causal model (i.e., the dashed line in Fig. 2 in the main paper). *It is worth noting that in this paper, we do not aim at discovering the latent confounders precisely.* Motivated by existing problems in NTL, we propose to disentangle the $C$ and $S$, thus avoiding the contradictions between models and human understanding.

**Assumptions about the statistical dependence between $C$ and $S$.** We only assume that the statistical dependence between $C$ and $S$ has a weak relation in the target domain. Thus, the $S$ in the target domain barely can predict $Y$, which motivates us to fit $S$ in the target domain during dual-path knowledge distillation. We do not need to assume the strength of the statistical dependence in the source domain, which means that the relation between $S$ and $C$ can be either strong or weak in the

source domain. In real world scenarios, such assumptions are always satisfied. Empirically, our experiments (Section 4 in the main paper) illustrate that by fitting $S$ in the target domain, the $f_{ntl}$ always has random classification accuracies. Thus, our assumptions are reasonable.

### E.2 Causal relationship between the style factor $S$ and the domain $D$

As shown in Fig. 14 (a), we mainly focus on the situation of $S$ causing $D$ in this paper (i.e., $S \to D$) (Huang et al., 2022; Liu et al., 2021; Mitrovic et al., 2021; Zhang et al., 2022). Besides, as shown in Fig. 14 (b), $D$ causing $S$ (i.e., $D \to S$) is also a popular situation considered by the community of domain adaptation and domain generalization (Kong et al., 2022; Liu et al., 2021; Lu et al., 2021; Lin et al., 2024). We argue that both directions are reasonable and do not contradict each other. In our definition, the complex *style factor* $S$ contains all other factors except the *content content* $C$. Parts of the *style factor* $S$ may depend on the sampling environment. If we determine the domain according to the environment, it belongs to $D \to S$. In contrast, sometimes we determine the domain by other parts of the *style factor* $S$ (e.g., whether containing human-added watermark or style augmentations). In these scenarios, assuming $S \to D$ is more appropriate. Thus, both directions are reasonable and may simultaneously exist between the *style factor* $S$ and the *domain* $D$.

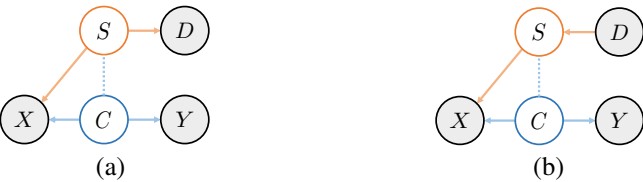

$$(a) \qquad\qquad\qquad (b)$$

Figure 14: Causal models with different causal direction between $S$ and $D$.

*More importantly, we do not focus on discovering the concrete causal directions between variables in this paper*. For completeness, we expand our H-NTL to $D \to S$ (we denote it as **H-NTL-D2S**). We illustrate the main principle of our H-NTL-D2S, based on which we conduct basic NTL experiments to show the effectiveness of H-NTL in the situation of $D \to S$.

**Main principle of the H-NTL-D2S.** In order to simultaneously infer *content factor* $C$ and *style factor* $S$ from sampled data $\mathcal{D}_s \cup \mathcal{D}_t$, according to the causal model in Fig. 14 (b), we factorize the joint distribution $P(X, Y, C, S, D)$ as follows:

$$P(X, Y, D, C, S) = P(D)P(C, S|D)P(Y|C, S)P(X|C, S). \tag{11}$$

Similar to the H-NTL focusing on $S$ to $D$ in the main paper, we also use VAE (Kingma & Welling, 2013) to infer latent factors $C$ and $S$. Specifically, we use two encoder modules $\hat{q}_{\phi_c}$ and $\hat{q}_{\phi_s}$ to model the posterior distributions $q_{\phi_c}(C|X)$ and $q_{\phi_s}(S|X)$ with learnable parameters $\phi_c$ and $\phi_s$, respectively. In addition, we use a decoder $\hat{p}_{\theta_x}$ to model the distribution $p_{\theta_x}(X|C, S)$ parameterized by $\theta_x$. To reduce computational costs and to allow our method efficiently infer latent factors, we approximate $P(C, S|D)$ by assuming that $X$ has sufficient information about $C$ while inferring $C$ from $S$ and $D$ is insignificant (Yao et al., 2021; Lu et al., 2018). Similarly, we assume $C$ contains sufficient information to predict the label $Y$. Thus, we use an encoder $\hat{p}_{\theta_s}$ to model the distributions $p_{\theta_s}(S|D)$ with learnable parameter $\theta_s$ and a classifier $\hat{p}_{\theta_y}$ to model the distributions $p_{\theta_y}(Y|C)$ with learnable parameter $\theta_y$. Totally, the VAE can be presented as: $\{\hat{q}_{\phi_c}, \hat{q}_{\phi_s}, \hat{p}_{\theta_x}, \hat{p}_{\theta_y}, \hat{p}_{\theta_s}\}$.

We follow the variational inference framework (Blei et al., 2017) to maximize the evidence lower-bound (ELBO) from the sampled observed data $(x, y, d) \in \mathcal{D}$. The ELBO$(x, y, d)$ is derived as:

$$
\begin{aligned}
\text{ELBO}(x, y, d) = &-\text{KL}(q_{\phi_c}(c|x)\|p(C)) - \text{KL}(q_{\phi_s}(s|x)\|p_{\theta_s}(s|d)) \\
&+ \mathbb{E}_{c \sim q_{\phi_c}(c|x)}\left[\log p_{\theta_y}(y|c)\right] \\
&+ \mathbb{E}_{c \sim q_{\phi_c}(c|x), s \sim q_{\phi_s}(s|x)}\left[\log p_{\theta_x}(x|c, s)\right].
\end{aligned} \tag{12}
$$

The detailed derivation of Eq. (12) is similar to Appendix B. We train the VAE by maximizing ELBO$(x, y, d)$ and thus obtain two encoders $\hat{q}_{\phi_c}(x)$ and $\hat{q}_{\phi_s}(x)$ for estimating unobservable *content factor* $C$ and *style factor* $S$, respectively. The remaining steps for non-transferable representation learning are the same as our main paper.

**Experiments.** As shown in Table 17, we present both target-specified and source-only NTL results for H-NTL-D2S. These results demonstrate that H-NTL-D2S can significantly degrade target domain performance and maintain source domain performance simultaneously, thus achieving the goal of NTL. In summary, our H-NTL is still effective when considering $D \to S$.

Table 17: Results of H-NTL-D2S on target-specified and source-only tasks. For each table cell, the first line shows averaged accuracy $Acc$ (%) with standard deviation, and the second line shows accuracy drop $\Delta$ (behind $\downarrow$) and relative drop $\Delta\%$ (in brackets) compared to supervised learning (SL).

| Source→ Target | Img Size | SL | | H-NTL-D2S (target-specified) | | H-NTL-D2S (source-only) | |
|---|---|---|---|---|---|---|---|
| | | Source | Target | Source | Target | Source | Target |
| MM→MT | 32 | 94.30 ±0.79 | 97.47 ±0.40 | 93.15 ±0.21 | 9.70 ±0.14 | 90.50 ±0.99 | 19.90 ±2.12 |
| | | – | – | ↓ 1.15 (1.22%) | ↓ 87.77 (90.05%) | ↓ 3.80 (4.03%) | ↓ 77.57 (79.58%) |
| SN→SD | 32 | 87.47 ±0.55 | 50.33 ±5.32 | 88.00 ±0.26 | 10.67 ±0.83 | 83.93 ±1.00 | 9.57 ±0.75 |
| | | – | – | ↓ -0.53 (-0.61%) | ↓ 39.66 (78.80%) | ↓ 3.54 (4.05%) | ↓ 40.76 (80.99%) |
| SD→MT | 32 | 98.23 ±0.06 | 55.30 ±3.00 | 97.13 ±0.32 | 11.13 ±3.11 | 91.77 ±1.56 | 12.80 ±7.57 |
| | | – | – | ↓ 1.1 (1.12%) | ↓ 44.17 (79.87%) | ↓ 6.46 (6.58%) | ↓ 42.50 (76.85%) |
| C10→S10 | 64 | 86.57 ±0.38 | 67.60 ±0.95 | 88.07 ±1.35 | 11.80 ±3.56 | 88.53 ±0.32 | 9.77 ±0.71 |
| | | – | – | ↓ -1.5 (-1.73%) | ↓ 55.80 (82.54%) | ↓ -1.96 (-2.26%) | ↓ 57.83 (85.55%) |
| VT→VV | 64 | 93.40 ±0.70 | 35.60 ±1.56 | 93.85 ±0.21 | 7.30 ±0.42 | 95.05 ±0.49 | 7.95 ±1.77 |
| | | – | – | ↓ -0.45 (-0.48%) | ↓ 28.30 (79.49%) | ↓ -1.65 (-1.77%) | ↓ 27.65 (77.67%) |
| OP→OC | 64 | 75.60 ±1.47 | 31.63 ±1.02 | 73.30 ±1.13 | 7.80 ±0.42 | 69.30 ±1.84 | 15.05 ±3.04 |
| | | – | – | ↓ 2.30 (3.04%) | ↓ 23.83 (75.34%) | ↓ 6.30 (8.33%) | ↓ 16.58 (52.42%) |

### E.3 THE MOTIVATION OF USING VARIATIONAL INFERENCE FRAMEWORK

We follow the variational inference framework (Blei et al., 2017) to model the **distribution** of latent variables $P(C, S)$ from the observed data, thus modeling the generative process. Other frameworks (such as auto-encoder (Wang et al., 2016)) cannot model the distribution. Moreover, there are also alternative options (for example, the VAEGAN (Xian et al., 2019; Chen et al., 2023b; Hong et al., 2022)) but with more complexity and may facing the problem of unstable training. Thus, the above factors are major motivations for us to use the variational inference framework.

## F LIMITATIONS AND FUTURE RESEARCH

In this section, we discuss the potential limitations of our work and areas for future research. *The major limitation of the proposed H-NTL is its training efficiency.* Due to the two-stage training processing (i.e., stage 1: disentanglement, stage 2: dual-path knowledge distillation), H-NTL needs more time and computation resources for training. Of course, the sacrifice of training time is worth it if we need better performance. Nevertheless, improving the training efficiency of H-NTL could still become a valuable research direction in the future. *However, it is worth noting that after being deployed in practice, our H-NTL will not have any disadvantages in computational cost.* In the inference phase, we only need the student model $f_{ntl}$ to predict. If the $f_{ntl}$ has the same network architecture as the network used in other NTL methods, H-NTL will occupy the same memory and have the same inference speed as other NTL methods. Moreover, because of the effective disentanglement and dual-path knowledge distillation, H-NTL allows a more lightweight student network to be taught to learn better non-transferable representations. This implies the potential advantages in the efficient inference of H-NTL in practical applications.

