# OpenReview forum: "Improving Non-Transferable Representation Learning by Harnessing Content and Style"
_ICLR.cc/2024/Conference — ICLR 2024 spotlight_

### Official Review · Reviewer_BF2B · 2023-10-23

**Soundness:** 3 good
**Presentation:** 3 good
**Contribution:** 3 good
**Rating:** 6
**Confidence:** 2

**Summary:**

The paper introduces a novel approach to non-transferable learning (NTL) that focuses on disentangling content and style factors, harnessing them as guidance for improved non-transferable representation learning. The concept of separating content and style for NTL is innovative and not commonly explored in existing literature. This approach presents a fresh perspective on NTL that has the potential to address the limitations of previous statistical methods. The novelty of this concept makes the paper stand out.

**Strengths:**

1. The paper is technically sound and demonstrates a comprehensive understanding of the issues in NTL. It effectively presents a causal model to explain non-transferable learning and provides a well-structured method to address these issues. The utilization of a variational inference framework for disentanglement and dual-path knowledge distillation for learning non-transferable representations is well-reasoned and technically sound.

2. The paper is well-organized and clearly written. It provides a coherent and logical progression from the problem statement to the proposed solution. The use of illustrative figures and the clear description of the causal model enhance the paper's overall clarity. However, the high complexity of the method and the underlying theoretical framework might make it challenging for some readers. To enhance clarity, it would be helpful to include examples or case studies illustrating the practical application of the method.

3. The paper is significant in the context of non-transferable learning. It offers a promising approach that addresses the limitations of previous methods, particularly the challenges related to statistical dependence between source and target domains. The disentanglement of content and style factors is an important contribution as it aligns with human-like understanding and, as demonstrated through experiments, improves NTL performance. The potential impact on the AI research community lies in its ability to advance the field of NTL, making it more effective and practical.

**Weaknesses:**

1. The paper is generally well-structured, but it could benefit from the inclusion of practical examples or use cases to demonstrate the application of the proposed method.

2. The experiments demonstrate the superiority of H-NTL over competing methods, which is a significant point in favor of the paper's claims. It would be helpful to include discussions on potential real-world applications where H-NTL could be particularly valuable.

3. The introduction could be more concise and direct, focusing on the problem and motivation for the proposed approach.

4. Minor grammatical and typographical errors should be addressed for a more polished final version.

5. Including discussions on potential limitations and areas for future research could provide a more well-rounded view of the work.

**Questions:**

see weakness

---

> ### Author Response · Authors · 2023-11-21
> **Response to Reviewer BF2B (1/3)**
>
> Thanks for your thoughtful reviews! We address your concerns as follows.
>
> > **Q1:It could benefit from the inclusion of practical examples or use cases to demonstrate the application of the proposed method.**  and
> > **Q2: It would be helpful to include discussions on potential real-world applications where H-NTL could be particularly valuable.**
>
> Thanks for your valuable suggestions. The practical application of the proposed H-NTL lies in the model intellectual property (IP) protections. Our H-NTL can provide two kinds of IP protection: ownership verification, and applicability authorization. As you suggested, we discuss and run more experiments on ownership verification and applicability authorization to further show the effectiveness of our H-NTL on real-world applications.
> - **Ownership verification**:
>   - The aim of ownership verification is to verify the ownership of a deep learning model. The proposed H-NTL provides a solution for ownership verification by triggering misclassification on the target domain [C1]. Specifically, we can add a pre-defined trigger patch (only known by the model owner) on the original dataset (i.e., the source domain) and see them as the target domain. It is worth noting that such a pre-defined trigger patch can be controlled to be shallow so that normal supervised learning (SL) models trained on the original source domain can still have normal performance on the patched source domain. Then, we train a deep learning model by using the proposed H-NTL on these two domains. After that, the training model will perform poorly on the data with the patch but have a good performance on the data without the patch. Thus, by observing the performance difference of a trained model on the source domain data with and without the pre-defined trigger patch, we can verify whether a deep learning model belongs to the model owner.
>   - We have verified the strong performance of the proposed H-NTL on watermark data with different patch values (results are shown in Section 4.1 and Figure 4 in our main paper), which serves as the core task of ownership verification.
>   - Here, we run additional experiments of ownership verification, and results are shown in **Table C1**. We can see that models trained with H-NTL behave differently on the data with and without the patch, whereas SL models perform nearly the same. Accordingly, the ownership of trained models can be verified.
> - **Applicability authorization**:
>   - Applicability authorization aims at authorizing models to certain data for preventing their usage on unauthorized data [C1], which can be solved by applying source-only H-NTL to restrict the model generalization ability to only the authorized domain. Specifically, we add a pre-defined authorized patch to the original data to be authorized and see them as the source domain. We regard *the union of the original data (without the authorized patch), the augmented original data with and without the authorized patch* as the target domain. Then, we train a deep learning model by using the proposed H-NTL on these two domains. After that, the trained model will only perform well on the authorized data (i.e., the original data with the authorized patch). For unauthorized data (e.g., the original data without the authorized patch, the data from other domains with or without the authorized patch), the trained model will perform poorly. Thus, we achieve the model applicability authorization.
>   - We run experiments of applicability authorization, and results are shown in **Table C2** (Digits datasets) and **Table C3** (complex datasets). We can see that the model trained by H-NTL has good performance on the authorized domain (highlighted in **bold** in each row) and poor performance on all unauthorized domains. This shows the effectiveness of H-NTL in applicability authorization.
>
> *All the discussion of practical applications and their experiments are added in Appendix D.2 of the revised paper (highlighted in purple). Thanks for your constructive suggestions!*

---

> ### Author Response · Authors · 2023-11-21
> **Response to Reviewer BF2B (2/3)**
>
> **Table C1:** Results of H-NTL on ownership verification. "(p)" means patched data. The results of H-NTL are highlighted in **bold**.
> | Source domain | Target domain | SL(patch/non-patch) | H-NTL(patch/non-patch) |
> | :-----------: | :-----------: | :-----------------: | :--------------------: |
> |      MT       |     MT(p)     |      98.8/98.5      |     **98.6/10.6**      |
> |      US       |     US(p)     |      98.9/98.8      |     **99.4/14.3**      |
> |      SN       |     SN(p)     |      87.9/87.4      |      **87.8/8.5**      |
> |      MM       |     MM(p)     |      92.8/92.0      |      **93.4/9.7**      |
> |      SD       |     SD(p)     |      96.3/95.9      |      **96.1/9.5**      |
> |      C10      |    C10(p)     |      86.5/60.2      |      **87.9/8.9**      |
> |      S10      |    S10(p)     |      88.1/82.0      |     **89.5/10.9**      |
> |      VT       |     VT(p)     |      93.4/92.2      |      **95.1/8.3**      |
>
> **Table C2:** Results of H-NTL on applicability authorization (digits datasets). "(p)" means data with the authorized patch. In each row, the authorized domain is highlighted in **bold**. The last column shows the averaged results on authorized and unauthorized domains (also highlighted in **bold**).
> | Source domain |  MT(p)   |  US(p)   |  SN(p)   |  MM(p)   |  SD(p)   |  MT   |  US   |   SN |  MM   |  SD   | H-NTL(auth/unauth) |
> | :-----------: | :------: | :------: | :------: | :------: | :------: | :---: | :---: | ---: | :---: | :---: | :----------------: |
> |     MT(p)     | **98.2** |   12.3   |   10.1   |   12.3   |   9.6    |  9.2  |  9.7  | 10.1 |  8.4  |  9.6  |   **98.2/10.1**    |
> |     US(p)     |   9.5    | **99.2** |   6.0    |   10.8   |   10.2   |  9.5  | 17.2  |  6.0 | 10.8  | 10.2  |    **99.2/8.9**    |
> |     SN(p)     |   16.5   |   18.8   | **88.3** |   11.9   |   10.5   |  8.9  |  9.2  | 17.0 | 10.3  | 10.0  |   **88.3/12.6**    |
> |     MM(p)     |   20.3   |   17.2   |   14.5   | **92.2** |   10.9   |  9.7  | 17.2  |  5.8 | 12.5  | 11.0  |   **92.2/13.2**    |
> |     SD(p)     |   8.7    |   29.5   |   22.0   |   17.1   | **94.4** |  8.1  |  7.5  |  9.3 | 10.9  | 11.9  |   **94.4/13.9**    |
>
> **Table C3:** Results of H-NTL on applicability authorization (complex datasets). "(p)" means data with the authorized patch. In each row, the authorized domain is highlighted in **bold**. The last column shows the averaged results on authorized and unauthorized domains (also highlighted in **bold**).
> |   Source domain   |  C10(p)   |  S10(p)   |  C10   |  S10   |   H-NTL(auth/unauth)   |
> | :---------------: | :-------: | :-------: | :----: | :----: | :--------------------: |
> |      C10(p)       | **87.4**  |    9.7    |  10.6  |  8.9   |      **87.4/9.7**      |
> |      S10(p)       |   12.7    | **88.3**  |  8.9   |  11.2  |     **88.3/10.9**      |
> | **Source domain** | **VT(p)** | **VV(p)** | **VT** | **VV** | **H-NTL(auth/unauth)** |
> |       VT(p)       | **92.0**  |    8.6    |  10.9  |  9.5   |      **92.0/9.7**      |

---

> ### Author Response · Authors · 2023-11-21
> **Response to Reviewer BF2B (3/3)**
>
> > **Q3: The introduction could be more concise and direct, focusing on the problem and motivation for the proposed approach.**
>
> Thanks for your helpful suggestion! We reorganize our introduction as follows:
> - The first paragraph describes the background of NTL.
> - The second paragraph analysis problems of existing methods, and the third paragraph further provides an example to intuitively illustrate the problems.
> - The remaining paragraphs contain the intuitive explanation for the proposed H-NTL method.
>
> *The revisions are highlighted in the updated manuscript (highlighted in purple).*
>
>
> > **Q4: Minor grammatical and typographical errors should be addressed for a more polished final version.**
>
> Thanks for pointing this out. We have tried our best to revise obvious typos/grammatical errors in the initial manuscript. We will continue to polish our manuscript to further improve our introduction and revise potential grammatical and typographical errors, thus making our final version more readable.
>
> > **Q5: Including discussions on potential limitations and areas for future research could provide a more well-rounded view of the work.**
>
> Thanks for your constructive suggestions. Here, we discuss the potential limitations of our work and areas for future research. *The major limitation of the proposed H-NTL is its training efficiency.* Due to the two-stage training processing (i.e., stage 1: disentanglement, stage 2: dual-path knowledge distillation), H-NTL needs more time and computation resources for training. Of course, the sacrifice of training time is worth it if we need better performance. Nevertheless, improving the training efficiency of H-NTL could still become a valuable research direction in the future. *However, it is worth noting that after being deployed in practice, our H-NTL will not have any disadvantages in computational cost.* In the inference phase, we only need the student model $f_{ntl}$ to predict. If the $f_{ntl}$ has the same network architecture as the network used in other NTL methods, H-NTL will occupy the same memory and have the same inference speed as other NTL methods. Moreover, because of the effective disentanglement and dual-path knowledge distillation, H-NTL allows a more lightweight student network to be taught to learn better non-transferable representations. This implies the potential advantages in the efficient inference of H-NTL in practical applications.
> *The discussion of limitations and future research is added in Appendix F of the revised paper (highlighted in purple). Thanks for your constructive suggestions!*
>
> ---
> [C1] Non-Transferable Learning: A New Approach for Model Ownership Verification and Applicability Authorization, ICLR 2022

---

### Official Review · Reviewer_fXnt · 2023-10-30

**Soundness:** 4 excellent
**Presentation:** 3 good
**Contribution:** 3 good
**Rating:** 8
**Confidence:** 3

**Summary:**

The paper proposes H-NTL, a new NTL method to address the issues of spurious correlation and fake independence present in many real-world implementations of NTL. The H-NTL method leverages a causal framework with two content and style latent factors to learn a variational inference framework. Empirical evaluations further support the strength of the proposed H-NTL method under various settings.

**Strengths:**

1. The paper is clearly written with presents a well-motivated justification for the H-NTL method.
2. Additional supplementary and ablation studies provide further evidence for the methodology.
3. H-NTL presents strong empirical performance in comparison with prior works.

**Weaknesses:**

No major weakness to note, however the reviewer would like to see additional evaluations with higher resolution images if possible (see questions below).

**Questions:**

Are there other standard NTL experimental setups using higher resolution datasets beyond the 32x32 or 64x64 images presented in the current paper?

---

> ### Author Response · Authors · 2023-11-21
> **Response to Reviewer fXnt**
>
> Thanks for your strong support and valuable comments! We address the weaknesses below:
>
> > **Q1: Are there other standard NTL experimental setups using higher resolution datasets beyond the 32x32 or 64x64 images presented in the current paper?**
>
> As you suggested, we run additional evaluations with higher-resolution images. In detail, we run experiments on the original resolution of VT$\rightarrow$VV (112$\times$112) and OP$\rightarrow$OC (224$\times$224) without resizing, thus showing the effectiveness of the proposed H-NTL on higher resolution images. Results of the proposed H-NTL on the target-specified NTL setting and source-only NTL setting are shown in **Table B1** and **Table B2**, respectively. On higher-resolution datasets, the proposed H-NTL can still effectively maintain source domain performance (comparable to SL) and significantly degrade the target domain performance.
> *We add the experiments of high-resolution data in Appendix D.1 of the revised paper (highlighted in purple). Thanks for your constructive suggestions!*
>
> **Table B1:** Results of **target-specified H-NTL** on high-resolution datasets.
> | Source domain | Target domain |  Image Resolution  | SL(source/target) | H-NTL(source/target) | source_drop | target_drop |
> | :-----------: | :-----------: | :----------------: | :---------------: | :------------------: | :---------: | :---------: |
> |      VT       |      VV       |    32$\times$32    |     89.7/22.4     |       91.7/8.1       |    -2.0     |    14.3     |
> |      VT       |      VV       |    64$\times$64    |     93.4/35.6     |       94.6/8.2       |    -1.2     |    27.4     |
> |      VT       |      VV       | **112$\times$112** |     97.5/42.6     |       97.8/8.3       |    -0.3     |    34.3     |
> |      OP       |      OC       |    32$\times$32    |     65.6/23.6     |       65.6/5.5       |     0.0     |    18.1     |
> |      OP       |      OC       |    64$\times$64    |     75.6/31.6     |       76.4/6.7       |    -0.8     |    24.9     |
> |      OP       |      OC       | **224$\times$224** |     86.0/35.5     |       84.9/3.6       |     1.1     |    31.9     |
>
> **Table B2:** Results of **source-only H-NTL** on high-resolution datasets.
> | Source domain | Target domain |  Image Resolution  | SL(source/target) | H-NTL(source/target) | source_drop | target_drop |
> | :-----------: | :-----------: | :----------------: | :---------------: | :------------------: | :---------: | :---------: |
> |      VT       |      VV       |    32$\times$32    |     89.7/22.4     |       91.6/9.3       |    -1.9     |    13.1     |
> |      VT       |      VV       |    64$\times$64    |     93.4/35.6     |       94.9/7.8       |    -1.5     |    27.8     |
> |      VT       |      VV       | **112$\times$112** |     97.5/42.6     |       96.7/8.2       |     0.8     |    34.4     |
> |      OP       |      OC       |    32$\times$32    |     65.6/23.6     |      63.7/14.4       |     1.9     |     9.2     |
> |      OP       |      OC       |    64$\times$64    |     75.6/31.6     |      71.6/16.0       |     4.0     |    15.6     |
> |      OP       |      OC       | **224$\times$224** |     86.0/35.5     |      83.3/10.1       |     2.7     |    25.4     |

---

> > ### Comment · Reviewer_fXnt · 2023-11-23
> >
> > Many thanks to the authors for responding to my questions.

---

### Official Review · Reviewer_AgKr · 2023-11-01

**Soundness:** 3 good
**Presentation:** 3 good
**Contribution:** 3 good
**Rating:** 6
**Confidence:** 3

**Summary:**

The authors explore an important problem named non-transferable learning, which aims to reduce the performance of a method on target domains while keeping considerable performance on source domains. The authors discover that existing methods suffer from two challenges, i.e., spurious correlation and fake independence. To deal with these challenges, the authors propose a variational inference framework that explicitly considers the contents and styles in various domains. The extensive experiments conducted on various datasets further demonstrate the effectiveness.

**Strengths:**

1. The paper is well-written and easy to follow.

2. The authors explore the important problem of non-transferable learning with two challenges investigated.

3. The authors conduct extensive experiments to showcase the superior performance of the work.

**Weaknesses:**

1. The authors state that existing methods suffer from the limitations of spurious correlation and fake independence. However, the authors do not provide any quantitative evaluation regarding these two challenges, except only intuitions.

2. The authors do not provide further details about the datasets used in the experiments. For example, the number of samples in each domain and each class. This will result in difficulties in understanding different datasets.

3. In Table 2 presented in the experimental part, the authors only consider two baselines for comparison, which is not fair enough for a comprehensive evaluation.

**Questions:**

Have the authors considered the benefits of using a variantiational inference framework? Is it due to the stochasticity?

---

> ### Author Response · Authors · 2023-11-21
> **Response to Reviewer AgKr (1/2)**
>
> Thanks for your helpful comments! We address your concerns as follows.
>
>
> > **Q1: The authors state that existing methods suffer from the limitations of spurious correlation and fake independence. However, the authors do not provide any quantitative evaluation regarding these two challenges, except only intuitions.**
>
> Thanks for pointing this out. It is difficult to directly quantify how many spurious correlations and fake independence the existing methods rely on in real-world datasets (it needs to manually analyze each sample and its prediction). Thus, we have validated the limitation of existing methods through a toy experiment (we present it in Appendix A due to the limited pages of the main paper).
>
> We leverage prior knowledge of spurious correlations to design a source domain, thus demonstrating the vulnerability (caused by fitting spurious correlation and modeling fake independence) of statistical-based methods. Specifically, we conduct the toy NTL experiment on digit datasets, where we use MNIST as the source domain and USPS as the target domain. Each digit in the source domain has a human-added class-wise color, thus creating a strong spurious correlation between the color and label. Our experiments show that the statistical-based method NTL [A1] and the proposed H-NTL achieve perfect non-transferable learning performance on the regular test data. However, when facing distribution shifts (e.g., perturbing the color), the statistical-based method NTL [A1] has serious performance degradation on the source domain, but the proposed H-NTL can still maintain the ideal performance. This phenomenon illustrates that the statistical-based method heavily relies on learning the spurious correlations to implement non-transferable learning. (More details please refer to Appendix A)
>
> > **Q2: The authors do not provide further details about the datasets used in the experiments. For example, the number of samples in each domain and each class. This will result in difficulties in understanding different datasets.**
>
> We sincerely apologize for the confusion. Due to the limited space of the main paper, we provided detailed information for each dataset used in our experiments in Appendix C.1 and the training details including the number of samples in each domain and each class in Appendix C.2.

---

> ### Author Response · Authors · 2023-11-21
> **Response to Reviewer AgKr (2/2)**
>
> > **Q3: In Table 2 presented in the experimental part, the authors only consider two baselines for comparison, which is not fair enough for a comprehensive evaluation.**
>
> Thanks for your constructive comments! Since non-transferable learning is a recently proposed problem and has few baselines, here we design an additional method (denoted as TargetClass) that can be used for non-transferable learning, thus serving as a baseline for comparison. Specifically, considering an N-class non-transferable learning task, we take the same backbone as our main paper as a feature extractor and use a classification head with *N+1 classes* to predict the N classes in the source domain, but predict an additional “target class" as class N+1 for all samples in the target domain. Therefore, the model trained by TargetClass will map all the target domain data to an isolated cluster that stays away from the N source-domain-class clusters in the feature space, thus hindering the source-to-target knowledge transfer.
> We run experiments under the same setting as our main paper, and results are shown in **Table A1** (on natural datasets) and **Table A2** (on watermark target-domain). From the comparison, the proposed H-NTL generally has better performance.  The baseline method TargetClass is usually hard to maintain the source domain performance and degrade the target domain performance (e.g., on OP$\rightarrow$OC 32$\times$32 and 64$\times$64, C10$\rightarrow$S10 32$\times$32 and 64$\times$64 in **Table A1**, and most watermark experiments in **Table A2**).
> *The comparison with the new baseline is added in Appendix D.3 of the revised paper (highlighted in purple). Thanks for your constructive suggestions!*
>
> **Table A1:** Comparison of the new baseline (denoted as TargetClass) and the proposed H-NTL on natural data. The best results are highlighted in **bold**.
> | Source domain | Target domain | Image Resolution | TargetClass(src/tgt) | H-NTL(src/tgt) |
> | :-----------: | :-----------: | :--------------: | :------------------: | :------------: |
> |      MM       |      MT       |   32$\times$32   |     **92.8/9.5**     |    93.1/9.9    |
> |      SN       |      SD       |   32$\times$32   |      87.8/10.4       |  **88.1/9.2**  |
> |      SD       |      MT       |   32$\times$32   |      97.1/12.6       | **97.1/11.0**  |
> |      C10      |      S10      |   32$\times$32   |      81.3/50.2       | **80.6/28.1**  |
> |      C10      |      S10      |   64$\times$64   |      85.8/22.0       |  **87.6/9.6**  |
> |      VV       |      VT       |   32$\times$32   |      91.4/10.9       |  **91.7/8.1**  |
> |      VV       |      VT       |   64$\times$64   |       94.2/7.9       |  **94.6/8.2**  |
> |      OP       |      OC       |   32$\times$32   |       64.2/8.6       |  **65.6/5.5**  |
> |      OP       |      OC       |   64$\times$64   |       72.2/6.6       |  **76.4/6.7**  |
>
> **Table A2:** Comparison of the new baseline (denoted as TargetClass) and the proposed H-NTL on watermarked data. The best results are highlighted in **bold**.
> | Source domain | Target domain | PatchValue | TargetClass(src/tgt) | H-NTL(src/tgt) |
> | :-----------: | :-----------: | :--------: | :------------------: | :------------: |
> |      C10      |    C10(p)     |     20     |      83.0/10.7       |  **87.8/7.9**  |
> |      C10      |    C10(p)     |     40     |      85.5/10.0       | **88.7/10.1**  |
> |      C10      |    C10(p)     |     60     |       85.5/8.8       |  **87.3/9.8**  |
> |      S10      |    S10(p)     |     20     |      85.4/53.4       | **83.6/16.7**  |
> |      S10      |    S10(p)     |     40     |      85.9/20.4       | **87.4/12.5**  |
> |      S10      |    S10(p)     |     60     |      86.9/15.9       |  **88.3/9.0**  |
>
>
>
> > **Q4: Have the authors considered the benefits of using a variational inference framework? Is it due to the stochasticity?**
>
>
> Thank you for the insightful question. Yes, you are right! More precisely, we need to model the **distribution** of latent variables $P(C,S)$ from the observed data, thus modeling the generative process. The variational inference framework helps us to model the distribution of latent variables $P(C,S)$. Other frameworks (such as auto-encoder) cannot model the distribution. Moreover, there are also alternative options (for example, the VAEGAN) but with more complexity and may facing the problem of unstable training. Thus, the above factors are major motivations for us to use the variational inference framework.
> *We add the discussion of the benefits of using a variational inference framework in Appendix E.3 (highlighted in purple in the revised manuscript).*
>
>
> ---
>
> [A1] Non-Transferable Learning: A New Approach for Model Ownership Verification and Applicability Authorization, ICLR 2022

---

> ### Comment · Reviewer_AgKr · 2023-11-23
>
> Thanks for your detailed response!

---

### Author Response · Authors · 2023-11-21
**General Response to All Reviewers**

We thank the reviewers for their insightful and constructive reviews of our manuscript. We are glad that all the reviewers (`AgKr`, `fXnt`, `BF2B`) found that: our paper is well-written and easy to follow; our paper is well-motivated and significant to NTL; and our method is reasonable and has good empirical performance.

Based on all the reviews, here we provide a general response to the concerns raised by multiple reviewers. The individual responses are commented on below each review.
- Regarding questions about experiments, we have addressed the concerns as follows:
  - We add a baseline method for comparison. (For Reviewer `AgKr`)
  - We run experiments on high-resolution images. (For Reviewer `fXnt`)
  - We run experiments on real applications (ownership verification and applicability authorization) (For Reviewer `BF2B`)
- Regarding questions about clarification for details, we have addressed the concern as follows:
  - We clarify our experiment setups (For Reviewer `AgKr`).
  - We add the potential limitations and future research in our paper. (For Reviewer `BF2B`)

We have also uploaded a revised version of our submission according to the suggestions provided by reviewers, with major changes highlighted in purple (there are also other minor changes in wording and typos). We look forward to further feedback and discussion. Please feel free to let us know if further details/explanations are helpful.

Best, Authors.

---

### Author Response · Authors · 2023-11-22
**Thanks**

Dear Reviewers,

Thank you very much for your efforts and constructive comments. Please note that the rolling discussion will end soon. If you have any further concerns or suggestions, kindly let us know. We are more than happy to address them and revise our paper accordingly.

Warm regards,

Authors

---

### Meta-Review · Area_Chair_Pdnb · 2023-12-06

**Metareview:**

The paper explores a problem of non transferable learning (NTL) aimed to restrict the generalizability of a model while maintaining its performance on the source domain. The NTL problem is relatively new, though some papers have dealt with it. The reviews agree the method provided here is novel, specifically separating content and style (see e.g., BF2B: “The concept of separating content and style for NTL is innovative and not commonly explored in existing literature. This approach presents a fresh perspective on NTL that has the potential to address the limitations of previous statistical methods.”). They also mention the paper is well written and easy to follow, and that it provides justification for the design choices. I agree with the conclusion of the reviews that this paper will be a welcome addition to ICLR and think that the innovative approach could help further the research related to NTL.

**Justification For Why Not Higher Score:**

The overall impact of the paper is limited. NTL is a new problem and isn't explored by a large set of researchers.

**Justification For Why Not Lower Score:**

The paper provides a novel approach to an area that already has publications. It would be interesting to the ICLR community.

---

### Decision · Program_Chairs · 2024-01-16

Accept (spotlight)